# PROPERTY-AWARE REINFORCEMENT LEARNING WITH RETRIEVAL ENHANCEMENT FOR CONTROLLABLE 3D MOLECULE GENERATION

## ABSTRACT

This paper studies the problem of controllable 3D molecule generation, which aims to design 3D molecules that satisfy given conditions. Previous methods usually incorporate the condition tokens into language models, and reconstruct molecules from the generated tokens. Despite their progress, performance remains unsatisfactory due to the neglect of conditional information during the generation process. To address this limitation, we propose a novel approach named Property-aware Reinforcement Learning with Retrieval Enhancement (POETIC) for controllable 3D molecule generation. To be specific, POETIC first tokenizes 3D molecular structures and leverages a language model (LM) for molecular generation. More importantly, it retrieves relevant samples with similar properties from an external database, which are used as prefixes to enhance generation quality. Furthermore, we pre-train a prediction model to identify the molecular properties, which in turn provides property-aware rewards for evaluation. These rewards guide reinforcement learning to optimize the LM. Extensive experiments on benchmark datasets validate the effectiveness of the proposed POETIC in comparison with state-of-the-art approaches. The source code is available at https://anonymous.4open.science/r/POETIC-BEA3.

## 1 INTRODUCTION

Molecule generation with desired properties is crucial for accelerating progress in drug discovery and materials science. Due to the vast size of chemical space, efficient generative models have emerged as a powerful framework for exploring and designing novel compounds (Vogt, 2023; Lavecchia, 2024). While early studies in 2D molecular graph generation demonstrated progress in validity and diversity (Jin et al., 2018; You et al., 2019; Kong et al., 2022), recent efforts increasingly focus on 3D molecule generation as three-dimensional structures govern molecular properties and are indispensable for structure-based drug design (Mansimov et al., 2019; Gebauer et al., 2020; Shi et al., 2021; Ganea et al., 2021; Satorras et al., 2022).

Building upon previous efforts in 3D molecule generation, recent advancements in 3D diffusion models have substantially improved the fidelity of 3D molecule generation by modeling atomic coordinates with strong equivariance (Hoogeboom et al., 2022; Xu et al., 2023; Morehead & Cheng, 2024). However, their reliance on long iterative denoising makes them computationally expensive and less attractive for scalable controllable design. In contrast, language model (LM) exploit geometry-aware tokenization and autoregressive decoding to enable more efficient generation and large-scale pretraining transfer (Li et al., 2024; Gao et al., 2024a). Despite these advantages, LMs are typically trained under maximum likelihood estimation (MLE), which prioritizes likelihood matching over goal-directed property optimization. This results in limited alignment with continuous molecular properties, and generalization further deteriorates when encountering out-of-vocabulary property values. Taken together, existing paradigms remain inadequate in achieving both precise controllability and robust generalizability, leaving a critical gap for subsequent exploration.

Prior works in language models (Devlin et al., 2019; Brown et al., 2020; Vaswani et al., 2023; Gu & Dao, 2024) suggest two complementary directions to address the issues we highlighted above (Gao et al., 2024b; Gupta et al., 2024; Wang et al., 2025d; Cao et al., 2024). On the

one hand, reinforcement learning (RL) has been widely used to align models with task-specific objectives, moving beyond distributional likelihood to enforce controllability (Schulman et al., 2017; Ouyang et al., 2022; Shao et al., 2024). On the other hand, retrieval-augmented generation (RAG) enhances generalization by grounding generation on external exemplars, as demonstrated in knowledge-intensive NLP tasks (Lewis et al., 2021; Borgeaud et al., 2022; Wang et al., 2025a;e).

Inspired by these insights, we conducted toy experiments to examine the effect of integrating RL and RAG into a molecular generator. Here, we use MLE to denote the baseline model trained solely with maximum likelihood estimation, without reinforcement learning or retrieval augmentation. As shown in Figure 1, RL substantially reduces error on in-distribution properties by enforcing property alignment, but its improvements come at the cost of weaker generalization. In contrast, RAG provides significant gains on unseen attributes, complementing RL by mitigating the out-of-vocabulary limitation.

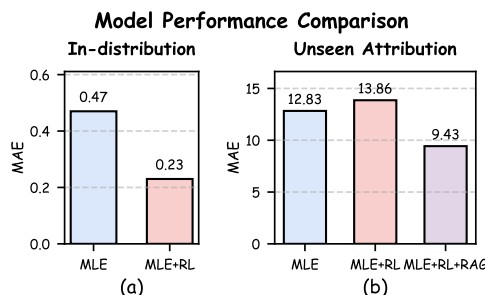

Figure 1: Toy experiment results.

Building on the above insights, we propose POETIC (Property-aware Reinforcement Learning with Retrieval Enhancement), a unified framework for controllable 3D molecule generation. POETIC employs a Mamba language model (Gu & Dao, 2024) as the generative backbone and integrates two complementary components: (i) *retrieval-augmented conditioning*, which retrieves property- and structure-similar exemplars from an external database and encodes them as compact prefixes to guide generation; and (ii) *property-aware reinforcement learning*, which leverages a frozen property predictor to provide explicit reward signals, ensuring alignment with continuous molecular targets. Extensive experiments on benchmark datasets demonstrate the effectiveness of our approach, which consistently outperforms state-of-the-art baselines by achieving controllability on in-distribution targets while maintaining robust generalization to unseen values.

In conclusion, the main contributions of our proposed POETIC can be highlighted as follows:

- **Novel Perspective.** To the best of our knowledge, this is the first framework that unifies retrieval and reinforcement learning for 3D controllable molecule generation using language models, directly addressing both controllability and generalizability.

- **Innovative Methodology.** POETIC integrates retrieval-augmented conditioning with property-aware reinforcement learning to enhance controllable generation, thereby improving generalizability to out-of-vocabulary properties and addressing the limitations of maximum likelihood training.

- **Empirical evaluation.** Comprehensive experiments demonstrate the effectiveness and generalization of POETIC for controllable 3D molecule generation, showing consistent improvements in target property control and extrapolation to unseen regions.

## 2 RELATED WORK

**3D Molecule Generation.** Diffusion models have become the dominant paradigm in 3D molecule generation, jointly modeling atom types and coordinate with E(3)-equivariant denoisers (Hoogeboom et al., 2022). To improve efficiency, latent-variable formulations have been introduced (Xu et al., 2023; You et al., 2024; Chen, 2024; Zhang et al., 2025), while geometry-complete and Clifford-equivariant architectures have pushed fidelity further (Morehead & Cheng, 2024; Liu et al., 2025a). Alternatively, flow matching methods offer a computationally more efficient approach by directly learning vector fields to map a prior to the data distribution, thus avoiding iterative denoising (Dunn & Koes, 2024; Cao et al., 2025; Geffner et al., 2025). Despite their success, diffusion-based frameworks remain computationally expensive due to iterative denoising. To overcome these limitations, language models have emerged as a faster and more scalable alternative (Flam-Shepherd et al., 2023; Flam-Shepherd & Aspuru-Guzik, 2023). Geo2Seq (Li et al., 2024) introduces geometry-informed tokenization for autoregressive generation, with extensions exploring 3D coordinate tokenization (Gao et al., 2024a) and large-scale molecular pretraining (Flam-Shepherd et al., 2022;

Figure 2: Overview of POETIC pipeline. POETIC consists of two key components for controllable molecule generation: property-guided molecule retrieval and property-aware reinforcement learning. In addition, the retrieval module integrates property-based pre-selection with hybrid property–structure filtering to further improve generation quality.

Pei et al., 2025; Wang et al., 2025b; Liu et al., 2025b). However, LM-based approaches still face challenges in continuous property modeling and task alignment.

**Retrieval-Augmented Generation.** RAG has recently been applied to molecular design as a way to inject external chemical knowledge during generation (Lewis et al., 2021; Zhong et al., 2025). In fragment-based RAG, retrieval of molecular fragment information from databases guides molecular generation and optimizes structural alignment (Lee et al., 2024; Phillips et al., 2025; Peng & Han[1], 2025). In diffusion models, retrieval of structural templates or reference molecules from databases is integrated into the denoising process to enhance structural fidelity (Huang et al., 2024; Xu et al., 2025; Wang et al., 2025c). While these methods show the potential of retrieval-enhanced molecular generation, they remain limited to static augmentation, and integrating RAG with LM-based molecular generation is still underexplored (Sharma, 2025; Brown et al., 2025; Cheng et al., 2025).

**Reinforcement Learning.** RL offers another avenue for aligning molecular generation with task-specific objectives (Dodds et al., 2024; He et al., 2024; Li et al., 2025). REINVENT (Olivecrona et al., 2017) first applies policy gradient to optimize SMILES-based generation for drug-likeness and binding affinity. Subsequent works extended this line to multi-objective optimization and synthetic feasibility (Wang & Zhu, 2024; Park et al., 2024; Zhang et al., 2024; Yuan et al., 2025). Graph-based RL further enables structure-level rewards (Telepov et al., 2024; Yang et al., 2024; Zhang, 2024), and docking-guided RL has been proposed for structure-based design (Jeon & Kim, 2020; Yang et al., 2021; Xiong et al., 2023; Danel et al., 2023). In contrast, RL has rarely been applied to LM-based molecular generation (Cao et al., 2025; Ahmed & Mohammed, 2025). The integration of property-aware reward modeling is especially underexplored, which motivates our approach.

## 3 THE PROPOSED POETIC

**Problem Definition.** In this work, we study the problem of controllable molecule generation in 3D space. A molecule with $n$ atoms is represented as a 3D point cloud $\mathcal{G} = \{(r_i, z_i)\}_{i=1}^n$, where $r_i \in \mathbb{R}^3$ denotes the 3D coordinate of the $i$-th atom and $z_i \in \mathbb{Z}$ denotes its corresponding atom type. We consider a molecule dataset $\mathcal{D} = \{(\mathcal{G}_j, s_j)\}_{j=1}^M$, where each molecule $\mathcal{G}_j$ is annotated with its property value $s_j$. Our objective is to learn a conditional generative model $p_\theta(\mathcal{G} \mid s^\star)$, which can generate molecules consistent with a target property $s^\star$.

### 3.1 FRAMEWORK OVERVIEW

As illustrated in Figure 2, the POETIC workflow is a multi-step process designed to improve controllability in 3D molecule generation. First, we tokenize the 3D molecules with atomic coordinates into discrete token sequences. To enrich the condition representation, we retrieve molecules with similar properties from an external database and encode their numeric property values along with structural statistics as structured prefixes. Subsequently, the LM generates candidate molecules conditioned on the exemplar prefixes. Finally, a pre-trained property prediction model evaluates the generated molecules and produces property-aware rewards, which are leveraged in a reinforcement learning procedure to fine-tune the LM and enforce alignment with the target properties.

## 3.2 TWO-STEP PROPERTY-GUIDED RETRIEVAL FOR PREFIX GENERATION

To address the challenge of retrieving molecules with similar properties and structural characteristics, a straightforward single-step approach often fails to balance breadth and accuracy. Inspired by prior retrieval-based molecule generation methods (Wang et al., 2023), we adopt a two-step strategy that decouples the problem into two complementary sub-tasks. In the first step, we perform efficient property-based filtering to generate a candidate set of molecules with similar properties. In the second step, we refine this candidate set by computing a weighted score that integrates both property and structural similarity, ensuring high-precision retrieval.

**Step 1: Property-based Pre-selection.** The first step aims to efficiently narrow the search space to molecules with properties similar to the query. We begin by filtering the molecule database based on their property values. Let the database consist of pairs $\{(\mathcal{G}_j, s_j)\}_{j=1}^{M}$, where $\mathcal{G}_j$ represents the molecular structure and $s_j$ denotes its property value. We first rank the molecules by their absolute deviation $d_j = |s_j - s^\star|$ from the target property $s^\star$, and then retain the top-$K_{\text{pool}}$ candidates:

$$\mathcal{P} = \{j \mid \operatorname{rank}_j(d_j) \leq K_{\text{pool}}\}. \tag{1}$$

This step ensures that the pool $\mathcal{P}$ consists of molecules whose properties closely align with the target, providing a high-precision subset of candidates.

**Step 2: Hybrid Property–Structure Filtering.** While the pool $\mathcal{P}$ from Step 1 contains molecules with similar properties, it may still exhibit structural heterogeneity. To further refine the selection, we incorporate structural information by computing structural embeddings for each molecule. Specifically, we define a structural embedding $\mathbf{f}_j$ for each molecule $\mathcal{G}_j$ as a concatenation of element frequencies and atomic distance histograms:

$$\mathbf{f}_j = [\operatorname{hist}_{\text{elem}}(\mathcal{G}_j), \operatorname{hist}_{\text{d}}(\mathcal{G}_j)], \tag{2}$$

where $\operatorname{hist}_{\text{elem}}$ represents the frequency of different elements in the molecule and $\operatorname{hist}_{\text{d}}$ is a smoothed histogram of interatomic distances. To capture the common structural characteristics of the candidate set, we compute a prototype vector $\bar{\mathbf{f}}$ by averaging the embeddings of the molecules in $\mathcal{P}$:

$$\bar{\mathbf{f}} = \frac{1}{|\mathcal{P}|} \sum_{j \in \mathcal{P}} \mathbf{f}_j. \tag{3}$$

Next, we re-score each candidate by combining its property and structural similarities:

$$\operatorname{score}(j) = \gamma \left( -|s_j - s^\star| \right) + (1 - \gamma) \cos\langle \mathbf{f}_j, \bar{\mathbf{f}} \rangle. \tag{4}$$

Here, $\gamma$ is a weight hyperparameter that balances the importance of property-based and structure-based similarities, and $\cos\langle \cdot, \cdot \rangle$ represents the cosine similarity between a molecule and the prototype vector. Finally, molecules are ranked by their scores, and the top-$K$ samples are chosen to constitute the exemplar set $\mathcal{N}$. This procedure is efficient and yields an exemplar set that is both property-consistent and structurally coherent, providing reliable guidance for the subsequent generation process.

**Prefix Construction.** From the final exemplar set $\mathcal{N}$, we extract key statistics, namely the normalized element frequencies and the most prominent distance peaks. These statistics are then serialized into a compact, structured prefix format with special delimiters, ensuring that the generated prefix conveys both property-targeted and structurally relevant context for the language model (LM). The prefix format is as follows:

$$\texttt{[COND\_START]}\ s^\star\ \texttt{[ELEM\_FREQ]}\ \hat{\pi}(e)\ \texttt{[D\_PEAK]}\ \{[\ell_r, h_r]\}_r\ \texttt{[RAG\_END]}, \tag{5}$$

where $s^\star$ represents the target property value, $\hat{\pi}(e)$ denotes the normalized element frequencies, and $\{[\ell_r, h_r]\}_r$ lists the most significant atomic distance peaks. For brevity, the explicit end delimiters (e.g., $\texttt{[COND\_END]}$, $\texttt{[ELEM\_FREQ\_END]}$, $\texttt{[D\_PEAK\_END]}$) are omitted here, though they are present in the actual serialized data to guarantee strict boundary definitions. The structured nature of this format allows for clear separation and easy interpretation of each element, enhancing the efficiency and accuracy of the language model in generating molecules with the desired properties and structural characteristics. For illustration purposes, several complete examples of constructed prefixes are provided in Appendix F.

### 3.3 PROPERTY-AWARE REINFORCEMENT LEARNING WITH BACKWARD GUIDANCE

We cast controllable molecule generation as sequence-level reinforcement learning. At time step $t$, the model observes a state given by the retrieval-augmented prefix and the previously generated tokens, and emits an action corresponding to the next token. A trajectory thus forms a complete molecule $\mathcal{G}$ after termination. Our goal is to learn a policy $\pi_\theta$ that maximizes expected property-aligned reward under the target specification $s^\star$:

$$\max_\theta \quad \mathbb{E}_{\mathcal{G}\sim\pi_\theta(\cdot|s^\star)}\big[\, R(\mathcal{G}, s^\star)\,\big]. \tag{6}$$

We build upon the GRPO framework (Shao et al., 2024) and extend it with a frozen property evaluator and a backward guidance mechanism for token-level credit assignment in molecular sequences.

**GRPO Objective.** For each conditioning query $q$ (target $s^\star$ with its RAG prefix), the policy $\pi_\theta$ generates $G$ candidate trajectories $\{o_i\}_{i=1}^G$ (full molecules). The training objective combines a clipped likelihood-ratio term with KL regularization against a frozen reference policy $\pi_{\text{ref}}$:

$$\mathcal{L}_{\text{GRPO}} = -\mathbb{E}\left[\min\left(\rho_{i,t}\hat{A}_{i,t}, \text{clip}(\rho_{i,t}, 1-\varepsilon, 1+\varepsilon)\hat{A}_{i,t}\right)\right]$$
$$+ \beta\mathbb{E}\left[\exp(\Delta\log\pi) - \Delta\log\pi - 1\right] \tag{7}$$

where $\rho_{i,t} = \frac{\pi_\theta(o_{i,t}|q,o_{i,<t})}{\pi_{\theta_{\text{old}}}(o_{i,t}|q,o_{i,<t})}$ is the token-level importance ratio, $\varepsilon$ is the clipping threshold, and

$$\Delta\log\pi = \log\pi_{\text{ref}}(o_{i,t} \mid q, o_{i,<t}) - \log\pi_\theta(o_{i,t} \mid q, o_{i,<t}) \tag{8}$$

provides an unbiased estimator of $D_{\text{KL}}(\pi_\theta\|\pi_{\text{ref}})$ via $\mathbb{E}[\exp(\Delta\log\pi) - \Delta\log\pi - 1]$. A key component is the advantage $\hat{A}_{i,t}$, which is constructed from group-relative rewards to avoid learning a separate value function.

**Reward Model.** The reward signal is provided by an external pretrained and frozen Equivariant Graph Neural Network (EGNN) (Satorras et al., 2021), which maps a generated molecule $\mathcal{G}_i$ to a property prediction $\hat{s}(\mathcal{G}_i)$. Keeping the evaluator fixed prevents reward drift during policy updates and yields a stable optimization target. Given the target $s^\star$, we define a smooth, scale-aware reward with validity penalty:

$$r_i = \exp\left(-\frac{|\hat{s}(\mathcal{G}_i)-s^\star|}{\sigma}\right) - \mathbf{1}\{\text{invalid}(\mathcal{G}_i)\}\cdot\lambda_{\text{inv}}, \tag{9}$$

where $\sigma$ controls tolerance to deviations and $\lambda_{\text{inv}} > 0$ penalizes structurally unreasonable configurations. This design encourages closeness to the target while maintaining structural validity.

**Group-relative Normalization.** GRPO forms advantages by contrasting candidates generated under the same condition. For each query $q$, we standardize rewards within its group of size $G$:

$$\tilde{r}_i = \frac{r_i - \mu_{\text{grp}}}{\sigma_{\text{grp}}}, \tag{10}$$

yielding group-relative signals that are robust to absolute reward scale and property units, where $\mu_{\text{grp}} = \frac{1}{G}\sum_{j=1}^G r_j$ and $\sigma_{\text{grp}} = \sqrt{\frac{1}{G}\sum_{j=1}^G(r_j - \mu_{\text{grp}})^2}$. These standardized scores serve as the scalar advantages for credit assignment:

$$\hat{A}_{i,t} \triangleq \tilde{r}_i. \tag{11}$$

**Backward Guidance.** Properties are only available after a complete molecule is generated, so intermediate rewards are sparse. Therefore, we propagate the trajectory-level signal back to the token level while masking out the conditioning tokens. Let $t_{\text{prefix}}$ be the index of the first generative token. We assign

$$\hat{A}_{i,t} = \begin{cases} \tilde{r}_i, & \text{if } t \geq t_{\text{prefix}}, \\ 0, & \text{otherwise}, \end{cases} \quad \text{and} \quad m_t = \mathbf{1}\{t \geq t_{\text{prefix}}\}, \tag{12}$$

so that gradients only flow through the learnable part of the sequence. To reduce length bias, we apply a normalization factor over effective tokens, defined as follows:

$$\hat{A}_{i,t}^{\text{eff}} = \frac{\hat{A}_{i,t}}{\max(1, T_i - t_{\text{prefix}})}, \tag{13}$$

where $T_i$ is the length of candidate $i$ (padding positions are masked by $m_t$). In practice, we implement this using a token mask and broadcast the group-normalized advantage across all valid generative positions.

---

**Algorithm 1** Training Algorithm of POETIC

---

    **First Stage: RAG-based Language Model Pretraining**
    **Input:** Geometric data $\{(\mathcal{G}_j, s_j)\}_{j=1}^{M}$
    **Step 1: Property Pre-selection**
1: Compute the absolute deviation of the molecule property: $d_j = |s_j - s^\star|$;
2: Select top-$K_{\text{pool}}$ candidates by Eq. 1 to construct the set $\mathcal{P}$;
    **Step 2: Property–Structure Filtering**
3: Calculate embeddings $\mathbf{f}_j$ via Eq. 2 for $j \in \mathcal{P}$;
4: Compute prototype embedding $\bar{\mathbf{f}}$ using Eq. 3;
5: Select top-$K$ exemplars by Eq. 4 to form $\mathcal{N}$, and serialize prefix $\mathcal{P}^\star$ by Eq. 5;
6: Pre-train the language model using Maximum Likelihood Estimation on the RAG-derived pre-
    fixes, enabling property-conditioned structure learning.
    **Second Stage: Property-aware RL Fine-tuning**
    **Input:** policy $\pi_\theta$, reference $\pi_{\text{ref}} \leftarrow \pi_\theta$, frozen EGNN $\hat{s}(\cdot)$
7: **for** each training step **do**
8:     Sample $G$ molecules from $\pi_\theta$ conditioned on $(s^\star, \mathcal{P}^\star)$;
9:     Compute rewards $r_i$ with EGNN using Eq. 9;
10:    Obtain the normalized $\tilde{r}_i$ within the group by Eq. 10;
11:    Assign token-level advantages via backward guidance by combining Eq. 12 and Eq. 13;
12:    Compute the importance ratios $\rho_{i,t}$, and calculate $\Delta \log \pi$ by Eq. 8;
13:    Derive the GRPO loss by Eq. 7;
14:    Update $\theta$ and periodically refresh $\pi_{\text{ref}} \leftarrow \pi_\theta$;
15: **end for**
16: **return** Trained policy $\pi_\theta$

---

## 3.4 Overall Workflow

Our framework unifies retrieval-augmented generation (RAG) and reinforcement learning (RL) for controllable 3D molecule generation, effectively bridging context-aware retrieval with reward-driven optimization. Given a target property $s^\star$, RAG retrieves exemplar molecules via property–structure filtering and encodes their statistics into a prefix, which provides chemically meaningful context for the language model (LM). This setup naturally guides the LM toward realistic outputs. The RL stage further enforces property alignment: a frozen pre-trained EGNN predicts molecular properties, rewards are normalized within candidate groups, and backward guidance propagates these signals to token-level actions. The detailed training pipeline of POETIC is presented in Algorithm 1.

**Model Training.** We train the model in two main stages: pretraining and fine-tuning, progressively building from contextual learning to reward optimization. (i) *Pretraining stage*: The language model is conditioned on RAG-derived prefixes, and supervised learning is applied under a negative log-likelihood (NLL) objective. This enables the model to learn the distribution of molecular structures conditioned on the property-specific context provided by the prefixes, establishing a strong foundation for subsequent refinements. (ii) *Fine-tuning stage*: We initialize $\pi_\theta$ via supervised fine-tuning and set $\pi_{\text{ref}} \leftarrow \pi_\theta$. At each iteration, candidate molecules are generated by decoding a batch of targets with RAG-derived prefixes. A frozen EGNN then predicts the properties of the generated molecules, providing rewards with validity penalties. These rewards are standardized and used to compute token-level importance ratios $\rho_{i,t}$, which are then propagated using backward guidance with length normalization. The model is updated using the clipped objective, incorporating a KL term to stabilize the training. The reference policy is periodically refreshed to maintain a meaningful KL anchor. Over successive iterations, this process guides the model towards generating valid molecules that align closely with the target properties, balancing structural validity with property alignment.

**Molecule Sampling.** To sample from a trained model, we begin by retrieving exemplar molecules for the target property and serializing them into a prefix. The language model then performs a standard autoregressive decoding under this prefix, generating atom tokens sequentially until reaching the stop token or maximum length. Unlike during training, the reward model is not queried and the policy is not updated. This ensures that inference is as efficient as standard LM decoding while producing molecules that remain valid and consistent with the specified property.

Table 1: Controllable generation performance on QM9, reported in terms of property MAE (lower is better). The best and second-best results are highlighted in bold and underline, respectively.

| Property (Units) | $\alpha$ (Bohr$^3$) | $\Delta\varepsilon$ (meV) | $\varepsilon_{\text{HOMO}}$ (meV) | $\varepsilon_{\text{LUMO}}$ (meV) | $\mu$ (D) | $C_v \left(\frac{\text{cal}}{\text{mol}} \text{K}\right)$ |
|---|---|---|---|---|---|---|
| Data | 0.10 | 64 | 39 | 36 | 0.043 | 0.040 |
| Random | 9.01 | 1470 | 645 | 1457 | 1.616 | 6.857 |
| $N_{\text{atoms}}$ | 3.86 | 866 | 426 | 813 | 1.053 | 1.971 |
| EDM | 2.76 | 655 | 356 | 584 | 1.111 | 1.101 |
| GEOLDM | 2.37 | 587 | 340 | 522 | 1.108 | 1.025 |
| NExT-Mol | 1.16 | 297 | 205 | 235 | 0.507 | 0.512 |
| Geo2Seq with Mamba | 0.46 | 98 | 57 | 71 | 0.164 | 0.275 |
| POETIC | **0.21** | **62** | **39** | **27** | **0.080** | **0.077** |

## 4 EXPERIMENTS

### 4.1 EXPERIMENTAL SETTINGS

**Dataset.** We adopt QM9 (Ramakrishnan et al., 2014) for controllable molecule generation. The QM9 dataset is one of the most widely used benchmarks in quantum chemistry and machine learning research. It contains over 134K stable molecules composed of C, H, O, N, and F atoms, with up to nine heavy atoms. Following (Anderson et al., 2019), we partition the dataset into 100K training, 18K validation, and 13K test samples. For the controllable generation experiments, we adopt the EDM protocol (Hoogeboom et al., 2022) and further split the training set into two equal halves of 50K samples each. An EGNN-based predictor (Satorras et al., 2021) is trained on the first half to learn property prediction, while the conditional generative model is trained on the second half.

**Evaluation metrics.** We evaluate our model performance on QM9 from complementary perspectives: (i) *Controllability*: Following (Hoogeboom et al., 2022), we report mean absolute error (MAE) on six quantum properties: polarizability ($\alpha$), HOMO–LUMO gap ($\Delta\epsilon$), dipole moment ($\mu$), heat capacity ($C_v$), orbital energy ($\epsilon_{\text{HOMO}}$), and $\epsilon_{\text{LUMO}}$, with property values predicted by the same pre-trained EGNN used during RL fine-tuning. (ii) *Generalizability*: We additionally evaluate on unseen properties (out-of-vocabulary target values not observed during training), focusing on two representative quantum properties, polarizability ($\alpha$) and HOMO–LUMO gap ($\Delta\epsilon$), and and report MAE to assess the model's ability to generalize beyond the training distribution (see Appendix D for the full unseen-target evaluation protocol).

**Baselines.** We compare POETIC with recent state-of-the-art methods in controllable generation, including diffusion-based models (EDM (Hoogeboom et al., 2022), GeoLDM (Xu et al., 2023)), a language modeling approach (Geo2Seq with Mamba (Li et al., 2024)), and the hybrid framework NExT-Mol (Liu et al., 2025b). We also include three dataset-driven references: (i) Data, (ii) Random, and (iii) $N_{\text{atoms}}$. Further details of these baselines are provided in Appendix D.

**Implementation Details.** We train a 16-layer Mamba model (hidden size 768, context length 200) on QM9 using AdamW with batch size 32 for 200 epochs. For retrieval-augmented generation, we adopt a two-step strategy: property-based pre-selection to retain the top-$K_{\text{pool}} = 40$ candidates, followed by hybrid similarity ranking to choose $K = 5$ exemplars. The supervised model is further fine-tuned with property-aware reinforcement learning (temperature 0.7, top-$k = 80$) to encourage valid and property-consistent generations. Experiments are conducted on two RTX 4090D GPUs, with pretraining taking $\sim$4 hours and fine-tuning $\sim$3 hours.

### 4.2 RESULTS

After describing the experimental setup, we now present the results, covering two complementary aspects: (i) performance of controllable generation and (ii) performance of generalizability.

**Performance of Controllable Generation.** Table 1 summarizes controllable generation performance on QM9. POETIC consistently outperforms all prior approaches across the six quantum properties, achieving the lowest error in every case. We observe a below-evaluator phenomenon: MAE on HOMO, LUMO, and $\Delta\varepsilon$ falls below the dataset-level average of the frozen EGNN evaluator. Additional ablations show that this effect contributes only marginally to the overall gains, indi-

cating that most improvements stem from our method rather than evaluator alignment (Appendix C). These results highlight the complementary benefits of our design: retrieval-augmented prefix generation enhances the integration of conditional signals during molecule generation, while property-aware RL fine-tuning enables precise alignment with target values. Together, these components lead to more accurate and controllable 3D molecule generation, demonstrating the clear advantage of POETIC over state-of-the-art baselines.

**Performance of Generalizability.** To further assess the generalizability of our framework, we evaluate its ability to generalize to out-of-vocabulary property values, focusing on two representative quantum properties: polarizability ($\alpha$) and HOMO–LUMO gap ($\Delta\epsilon$). As shown in Table 2, our method outperforms the strongest baseline on both properties. These results indicate that

Table 2: Generalization performance on out-of-vocabulary properties, reported in MAE (lower is better, best in bold).

| **Property (Units)** | $\alpha$ (Bohr$^3$) | $\Delta\varepsilon$ (meV) |
|---|---|---|
| Geo2Seq with mamba | 14.67 | 2139 |
| POETIC | **9.24** | **1685** |

the proposed framework not only achieves strong controllability across six in-distribution properties, but also transfers effectively to unseen targets. The gains primarily stem from two complementary components: the retrieval-augmented prefix, which enriches the generative context with structural priors, and the property-aware reinforcement learning strategy, which provides explicit signals for navigating beyond the training distribution.

## 4.3 ABLATION STUDIES

To study the effects of retrieval-augmented generation (RAG) and reinforcement learning (RL) in POETIC, we design four variants: (i) maximum-likelihood language model (MLE), (ii) reinforcement learning fine-tuning (MLE + RL), (iii) retrieval-augmented conditioning (RAG + MLE), and (iv) the full model (POETIC) that combines RAG and RL. These variants allow us to isolate the role of each component and their combined impact on controllability and generalization. All variants are trained under the same setup as the main experiments in Section 4.1, and evaluated with the same metrics (in-distribution and unseen-property MAE). Additional ablations are provided in Appendix C, highlighting the role of structural priors in RAG and the contribution of RL.

**Ablation Results.** As shown in Table 3, the four variants highlight the complementary effects of RL and RAG. (i) For RL, introducing property-alignment rewards that directly supervise continuous property values leads to a substantial reduction in in-distribution MAE, demonstrating improved controllability. However, because RL primarily reinforces alignment on seen tokens, it weakens generalization and yields higher errors on unseen properties. (ii) By contrast, RAG con-

Table 3: Ablation study of RAG and RL in POETIC on QM9. Metrics: In-dist. (controllability) and Unseen (generalizability).

| **Model Variant** | In-dist. | Unseen |
|---|---|---|
| MLE | 1.06 | 14.44 |
| MLE + RL | 0.34 | 14.89 |
| RAG + MLE | 0.96 | 9.81 |
| POETIC | 0.21 | 9.24 |

ditions generation on retrieved neighbors that serve as structural priors, enriching the context and transferring useful patterns even for out-of-vocabulary tokens. This notably reduces unseen-property MAE and strengthens generalization, though its implicit and context-driven supervision is less effective for precise alignment within the training distribution. (iii) Integrating both components in the full POETIC model achieves the best balance: RL provides fine-grained controllability in-distribution, while RAG extends coverage to unseen tokens, keeping the unseen-property MAE competitive. Overall, this combination confirms that continuous reward optimization and retrieval-based structural priors address complementary weaknesses of maximum-likelihood training, together achieving both strong controllability and robust generalization.

## 4.4 HYPERPARAMETER SENSITIVITY ANALYSIS

In this section, we systematically evaluate the sensitivity of POETIC to key hyperparameters to assess its robustness. Experiments are conducted on the QM9 dataset, using a subset of 2,000 target property values for computational efficiency. As shown in Figure 3, POETIC remains stable across a broad range of settings. (i) **Retrieval parameters $K$ and $K_{\text{pool}}$:** Increasing $K$ (with $K_{\text{pool}}$ fixed) initially improves controllability (optimal around 3–7), but larger values introduce noisy exemplars

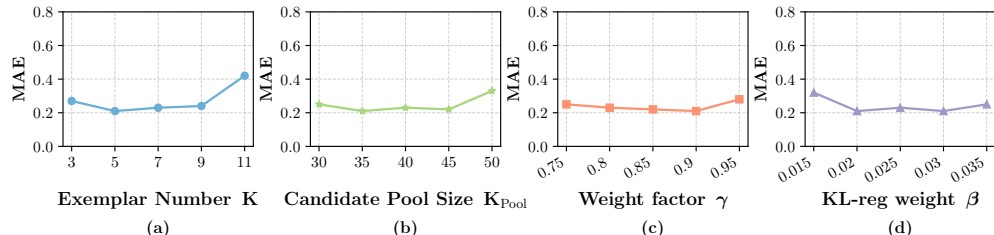

Figure 3: Hyperparameter Sensitivity Results. (a) shows the effect of exemplar number $K$, while (b) examines the candidate pool size $K_{\text{pool}}$. (c) describes the weighting factor $\gamma$ balancing property and structural similarity, and (d) presents results under different KL-regularization weights $\beta$.

and degrade performance. In contrast, varying $K_{\text{pool}}$ with $K$ fixed yields nearly unchanged results, indicating insensitivity to pool size. (ii) **Weighting factor $\gamma$:** Adjusting $\gamma$ to emphasize property over structural similarity leads to only minor changes. Importantly, structural similarity remains essential: property similarity drives the initial candidate selection, while structural cues in the second stage ensure chemically meaningful and diverse exemplars. (iii) **Regularization strength $\beta$:** Varying the KL-regularization weight $\beta$ leads to moderate performance differences but no drastic fluctuations, indicating that the model remains robust across the tested range. In summary, POETIC exhibits stable performance under diverse configurations, underscoring its robustness while highlighting the complementary roles of property and structural similarity in the two-stage retrieval framework, providing strong evidence of its practical reliability and consistency.

## 4.5 VISUALIZATION RESULTS

In this section, we visualize molecules generated by our POETIC conditioned on polarizability ($\alpha$). Polarizability characterizes how easily a molecule develops an electric dipole moment under an external field, and is closely linked to molecular size and structural flexibility.

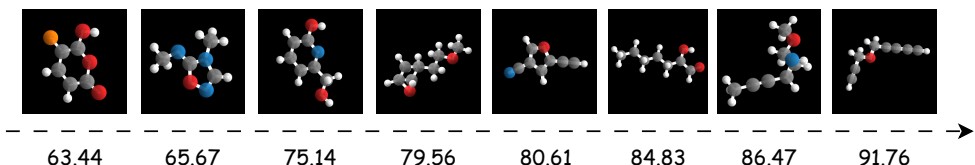

Figure 4: Examples of molecules generated by POETIC under varying polarizability $\alpha$, with the corresponding $\alpha$ shown below each image.

As shown in Figure 4, the generated molecules vary systematically with $\alpha$: higher values correspond to more extended, less symmetric conformations, consistent with the intuition that highly polarizable molecules deviate from compact geometries. These results demonstrate that our model captures meaningful structure–property relationships aligned with patterns observed in the QM9 dataset.

## 5 CONCLUSION

In this work, we introduce a novel reinforcement learning framework POETIC, which augmented with property-aware retrieval to tackle the challenge of controllable 3D molecule generation. To enhance generation quality, POETIC retrieves property-similar and structurally consistent molecules from an external database and further incorporates them as structured prefixes to the LM, thereby improving its generalization to unseen properties. To ensure property alignment, POETIC employs a frozen prediction model to evaluate candidate molecules and then generate property-aware rewards, which are subsequently leveraged to optimize the language model through reinforcement learning. This process encourages reliable 3D molecule generation conditioned on specific properties. Extensive experiments on benchmark datasets demonstrate that POETIC achieves superior performance and strong generalizability compared with state-of-the-art baselines. These above results highlight POETIC as a promising approach for 3D molecule generation with broad applicability across diverse scientific domains.

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

# A  LLM USAGE STATEMENT

A large language model was used solely to aid and polish the writing of this manuscript. Its assistance was limited to grammar correction, sentence-level rephrasing and minor improvements in clarity and readability. All scientific content, including the research ideas, experimental design, data analysis, and conclusions, was fully developed and verified by the authors. No text was automatically generated beyond these language refinements, and every suggested edit was manually reviewed before inclusion.

# B  ALGORITHMIC DETAILS

In the main text (Algorithm 1), we present the POETIC training pipeline as a unified procedure. For completeness, here we decompose it into two core modules: (i) the retrieval stage, which constructs property-guided prefixes, and (ii) the reinforcement learning stage, which fine-tunes the policy with property-aware rewards. This separation highlights how each component contributes individually to the overall framework.

---

**Algorithm 2** Two-step Property-Guided Retrieval for Prefix Generation

---

**Require:** Target $s^\star$, database $\{(\mathcal{G}_j, s_j)\}_{j=1}^M$, hyperparameters $K_{\text{pool}}, K, \gamma$
**Ensure:** Prefix $\mathcal{P}^\star$
  1: Compute property deviation $d_j = |s_j - s^\star|$ for all $j$
  2: Form candidate pool $\mathcal{P}$ with top-$K_{\text{pool}}$ entries by ascending $d_j$
  3: For $j \in \mathcal{P}$, compute $\mathbf{f}_j = [\text{hist}_{\text{elem}}(\mathcal{G}_j), \text{hist}_d(\mathcal{G}_j)]$
  4: Compute prototype $\bar{\mathbf{f}} = \frac{1}{|\mathcal{P}|} \sum_{j \in \mathcal{P}} \mathbf{f}_j$
  5: Re-score each $j \in \mathcal{P}$ with $\text{score}(j) = \gamma \cdot \big( - |s_j - s^\star| \big) + (1 - \gamma) \cdot \cos\langle \mathbf{f}_j, \bar{\mathbf{f}} \rangle$
  6: Select top-$K$ candidates by $\text{score}(\cdot)$ to form exemplar set $\mathcal{N}$
  7: Aggregate element frequencies and distance peaks from $\mathcal{N}$
  8: Serialize statistics into prefix $\mathcal{P}^\star$ and prepend to LM input

---

**Retrieval Stage: Two-step Property-guided Retrieval.** The retrieval component is summarized in Algorithm 2, which ensures that the language model is conditioned on exemplars that are both property-consistent and structurally coherent. This two-step process This process first performs a coarse property-based pre-selection, and then applies a hybrid filtering step that integrates both property and structural similarities. The resulting exemplar set is summarized into a compact prefix $\mathcal{P}^\star$, providing informative context for downstream generation.

---

**Algorithm 3** Property-aware GRPO with Backward Guidance

---

**Require:** Policy $\pi_\theta$, reference $\pi_{\text{ref}} \leftarrow \pi_\theta$, frozen EGNN $\hat{s}(\cdot)$, group size $G$, clip $\varepsilon$, KL weight $\beta$
  1: **for** each training step **do**
  2:     Sample $G$ candidates $\{\mathcal{G}_i\}$ from $\pi_\theta$
  3:     Compute rewards $r_i = \exp(-|\hat{s}(\mathcal{G}_i) - s^\star|/\sigma) - \mathbf{1}\{\text{invalid}(\mathcal{G}_i)\}\lambda_{\text{inv}}$
  4:     Normalize within group: $\tilde{r}_i = (r_i - \mu_{\text{grp}})/(\sigma_{\text{grp}})$
  5:     Set token-level advantages $\hat{A}_{i,t}$ via backward guidance
  6:     Compute ratio $r_{i,t} = \pi_\theta/\pi_{\theta_{\text{old}}}$ and loss

$$\mathcal{L} = -\mathbb{E}[\min(r_{i,t}\hat{A}_{i,t}, \text{clip}(r_{i,l}, 1 - \varepsilon, 1 + \varepsilon)\hat{A}_{i,t})] + \beta D_{\text{KL}}(\pi_\theta \| \pi_{\text{ref}})$$

  7:     Update $\theta$; periodically refresh $\pi_{\text{ref}} \leftarrow \pi_\theta$
  8: **end for**

---

**Reinforcement Learning Stage: Property-aware GRPO.** Algorithm 3 outlines the reinforcement learning procedure, where a frozen EGNN serves as a reward model to guide the policy towards the target property. Rewards are normalized within groups to reduce variance, and a backward guidance mechanism propagates sequence-level signals to token-level advantages. These components together stabilize optimization and enforce property alignment during training.

# C  ADDITIONAL ABLATION ANALYSIS OF HOMO, LUMO, AND THE HOMO–LUMO GAP

Among all target properties in QM9, the frontier orbital energies (HOMO and LUMO) are fundamental descriptors of molecular reactivity and stability. Their difference defines the HOMO–LUMO gap, $\Delta\epsilon = \epsilon_{\text{LUMO}} - \epsilon_{\text{HOMO}}$. Because the gap is a deterministic function of the two orbital energies, aligning generated molecules with respect to HOMO and LUMO naturally improves the alignment of $\Delta\epsilon$ as well (as measured by the EGNN evaluator). To investigate why these properties are particularly sensitive, we compare the full model against four targeted ablations: (i) **No-RL**, which removes reinforcement learning while keeping retrieval, testing whether retrieval alone suffices for controllability; (ii) **No-RAG**, which removes the retrieval module while retaining RL, isolating the role of property-aware retrieval; (iii) **Structure-ablated**, which further discards structural descriptors from the retrieved prefixes to assess the necessity of explicit structural cues; and (iv) **Prefix-randomized**, which preserves the retrieval size but shuffles prefix statistics to disrupt meaningful signals. The full (POETIC) model serves as the unablated reference. Taken together, these controlled experiments reveal that reinforcement learning provides the primary source of error reduction, while structural priors supplied through retrieval further refine property alignment, and both are necessary for precise regulation of frontier orbital properties.

Table 4: Ablation results on HOMO, LUMO, and the HOMO–LUMO gap (in-distribution MAE, meV).

| Model Variant | $\Delta\varepsilon$ (meV) | $\varepsilon_{\text{HOMO}}$ (meV) | $\varepsilon_{\text{LUMO}}$ (meV) |
|---|---|---|---|
| Full (POETIC) | **62** | **39** | **27** |
| No-RL | 151 | 78 | 94 |
| No-RAG | 76 | 45 | 32 |
| Structure-ablated | 78 | 47 | 35 |
| Prefix-randomized | 1033 | 998 | 883 |

The ablation results clarify why the errors of HOMO, LUMO, and their gap fall below the dataset-level average of the EGNN evaluator. Reinforcement learning provides the dominant optimization signal, while retrieval supplies structural priors that bias generation toward regions of chemical space where the evaluator is more stable. Together, these factors reduce the conditional error below the dataset-level baseline. Within this trend, the HOMO–LUMO gap shows the greatest improvement, consistent with prior studies highlighting its strong sensitivity to medium-range motifs such as conjugation length, aromatic topology, and heteroatom placement (Ramakrishnan et al., 2014). HOMO and LUMO also shift systematically with these motifs, though to a lesser extent, which matches the ablation results showing that all three properties depend critically on structural priors.

Reinforcement learning thus serves as the essential fine-tuning mechanism, translating deviations in the evaluated properties into consistent feedback and propagating this signal throughout the sequence. Retrieval complements this process by anchoring optimization in chemically plausible regions of molecular space, allowing RL to sharpen property alignment more effectively. Without RL, retrieval alone cannot reduce errors below the dataset average; without retrieval, RL loses the structural signals needed to fully exploit evaluator stability. Their combination is therefore crucial for achieving the observed improvements.

Finally, the fact that both the reward model used for RL and the evaluator share the same EGNN architecture may introduce a small compatibility gain. However, the ablation results show that this effect is limited: if architectural overlap were the dominant factor, the No-RAG variant (with RL but without retrieval) would already match the performance of the full model, yet its errors remain substantially higher. This indicates that while scorer–policy consistency may provide a minor boost, the major improvements stem from RL, with structural priors from retrieval enabling further reductions. Thus, the drop of HOMO, LUMO, and gap errors below the dataset-level EGNN error is best explained by the combined action of RL and retrieval, rather than by architectural overlap between the reward model and the evaluator.

# D EXPERIMENTAL DETAILS

## D.1 HYPERPARAMETERS AND EXPERIMENTAL DETAILS

**Supervised Pre-Training.** We train a 16-layer Mamba model with a hidden size of 768 on the QM9 dataset. The batch size is set to 32 and the base learning rate to $6 \times 10^{-4}$. Training runs for 200 epochs using the AdamW optimizer with a linear warmup followed by cosine decay: the learning rate increases linearly from zero to $6 \times 10^{-4}$ during the first 10% of training tokens, and then decays to $6 \times 10^{-4}$ according to a cosine schedule. Real numbers are tokenized to two decimal places, and the context length is fixed to 200 based on dataset statistics. All supervised experiments are conducted on two NVIDIA RTX 4090D GPUs.

For retrieval-augmented generation (RAG), we adopt a two-step retrieval strategy over QM9 molecular sequences. First, property-based pre-selection ranks molecules by their absolute deviation from the target property, and the top-$K_{\text{pool}} = 40$ candidates are retained. Second, the candidate set is refined with a hybrid similarity score that combines property alignment (weight 0.9) with sequence-level structural coherence (weight 0.1), from which the top-5 exemplars are selected. From these exemplars, we extract element frequency distributions and distance histogram peaks, and serialize them into structured prefixes with property values formatted to two-decimal precision.

**Reinforcement Learning Fine-tuning.** To further enforce alignment with target molecular properties, we fine-tune the supervised model using property-aware reinforcement learning. The setup employs a 16-layer Mamba with hidden size 768, context length 200, temperature 0.7, and top-$k$ sampling ($k = 80$). Each training step samples 16 conditions, with 12 generations per condition. Invalid or unparsable molecules receive a reward of $-1.0$, while valid generations are scored according to property objectives, with all rewards scaled by 1.0. Optimization uses AdamW with a learning rate of $1.2 \times 10^{-5}$, no weight decay, and gradient clipping of 1.0. Reinforcement learning runs for 800 iterations with distributed training on two NVIDIA RTX 4090D GPUs.

## D.2 BASELINES

We evaluate POETIC against a set of recent controllable molecular generation methods as well as several dataset-driven references. All baselines are trained and tested under the same benchmark setup as in our main experiments. Below we briefly describe each method.

**Model-based baselines.**

- EDM (Hoogeboom et al., 2022): EDM is an equivariant diffusion model that jointly generates discrete atom types and continuous 3D coordinates. It employs an E(3)-equivariant graph neural network (Satorras et al., 2021) to model the denoising process, and supports conditional generation by concatenating target molecular properties into node features.

- GeoLDM (Xu et al., 2023): GeoLDM first encodes molecules into a structured latent space that contains both equivariant tensor channels and invariant scalars. A latent diffusion model is then applied in this compact space, and a decoder reconstructs the 3D coordinates and atom types. Conditional generation is enabled by injecting property guidance in the latent diffusion stage.

- Geo2Seq with Mamba (Li et al., 2024): Geo2Seq discretizes molecular geometry into a canonical sequence using SE(3)-invariant spherical coordinates and a canonical labeling scheme. The sequence is modeled autoregressively with a Mamba-based language model. The generated sequence can be deterministically mapped back to a valid 3D structure, which allows efficient and geometry-aware controllable generation.

- NExT-Mol (Liu et al., 2025b): NExT-Mol combines diffusion modeling with sequence/latent-space representations to improve controllability and sampling efficiency. It represents a hybrid design that unifies different generative paradigms and serves as a strong recent baseline.

**Data-driven references.**

- Data: Uses the ground-truth QM9 labels as conditions. This captures the inherent error of the downstream property predictor and therefore serves as a reference lower bound.

- Random: Randomly shuffles molecular labels before conditioning. This breaks the true structure–property relation and serves as a reference upper bound for uncontrolled generation.

- $N_{\text{atoms}}$: Conditions only on the atom count of each molecule. This checks how much coarse-grained information (such as molecular size) alone can explain the predictive performance.

## D.3 UNSEEN-TARGET EVALUATION PROTOCOL

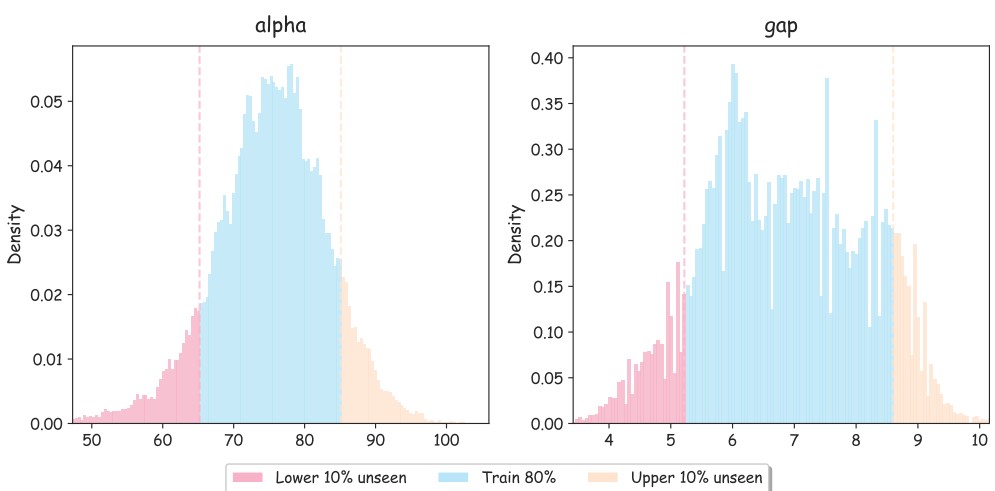

Figure 5: Distribution of molecular properties with the training region (middle 80%) and unseen regions (lower/upper 10%) highlighted.

To evaluate generalization beyond the training distribution, we further partition the training set by property values. For each property (e.g., polarizability $\alpha$ or HOMO–LUMO gap), we sort the values and determine the 10th and 90th percentiles. The central 80% interval is retained for model training, while the lower 10% and upper 10% tails are withheld and designated as *unseen targets*, as illustrated in Figure 5.

During evaluation, the model is conditioned on these unseen target values to generate molecular structures. Property values of the generated molecules are then predicted by the frozen property predictor described in the main text, and the mean absolute error (MAE) between generated and desired targets is reported. This protocol directly measures the model's ability to generalize controllable generation to property ranges that were never observed during training.

## D.4 DETAILS OF TOY EXPERIMENTS

To support the motivation in Section 1, we provide the configuration of the toy experiments shown in Figure 1. All experiments use the same model architecture and training setup as in the main results, but for efficiency we ran RL fine-tuning for 200 steps and evaluated on 200 randomly sampled target property values. We compared the following variants: MLE, trained by maximum likelihood without retrieval or reinforcement learning; MLE+RL, which augments MLE with property-aware reinforcement learning; and MLE+RAG+RL, which further incorporates retrieval-based structural prefixes. For the in-distribution case, targets were sampled from in-vocabulary property tokens, while for the unseen property case, targets were sampled from out-of-vocabulary property tokens.

## D.5 HYPERPARAMETER SENSITIVITY ANALYSIS SETTINGS

We provide the detailed settings for the hyperparameter sensitivity experiments reported in Section 4.4. All studies are conducted on the QM9 dataset using a randomly sampled subset of 2,000 target property values for computational efficiency. We vary three key factors: (i) the number of retrieved exemplars $K$ and the candidate pool size $K_{\text{pool}}$ in the two-step retrieval module; (ii) the weighting factor $\gamma$ that balances property and structural similarity during hybrid filtering; and (iii) the KL-regularization weight $\beta$ in the GRPO objective for reinforcement learning.

Unless otherwise stated, the default configuration follows the main experiments: $K = 5$, $K_{\text{pool}} = 40$, $\gamma = 0.9$, and $\beta = 0.03$. For each setting, we generate molecules conditioned on the same set of 2,000 target properties, evaluate property MAE with the frozen EGNN predictor, and keep all other model, training, and inference hyperparameters the same as in the main experiments.

# E  EXTENDED STUDIES

## E.1  NOVELTY AND VALIDITY EVALUATION

Table 5: Validity and novelty of molecules generated by POETIC across six quantum properties.

| Property | Validity (%) | Novelty (%) |
|---|---|---|
| $\alpha$ | 92.7 | 80.5 |
| $\Delta\epsilon$ | 90.3 | 70.1 |
| $\epsilon_{\text{HOMO}}$ | 87.7 | 76.7 |
| $\epsilon_{\text{LUMO}}$ | 91.4 | 81.8 |
| $\mu$ | 92.2 | 81.8 |
| $C_v$ | 94.4 | 76.6 |

We extend our evaluation on the QM9 dataset by introducing two additional metrics: validity and novelty. Following the methods from JODO (Huang et al., 2023) and Geo2Seq (Li et al., 2024), we use RDKit to convert 3D molecular structures into 2D graphs and evaluate the metrics on these 2D representations. The same model and experimental settings as in the main paper of POETIC are used. Table 5 summarizes the results across the six quantum properties. We observe that POETIC consistently achieves high validity, indicating that the model generates chemically valid and meaningful molecules. This confirms that the generated molecules are not only structurally feasible but also conform to expected chemical principles. Additionally, the model demonstrates a strong performance in novelty, as it produces a significant percentage of new molecules that are distinct from those seen during training. This highlights the model's ability to generate genuinely new molecular structures, rather than merely memorizing training data, further proving its creative potential in molecular design.

## E.2  ERROR CASE ANALYSIS

One common challenge in LM-based molecular generation is the emergence of hallucinations and repetitions. Prior work such as Geo2Seq (Li et al., 2024) has shown that autoregressive decoding without structural constraints can lead to repetitive token loops or to hallucinated geometries. In our supervised-only setting, we observed similar issues: the model occasionally repeated tokens excessively or generated invalid structures that broke chemical rules.

- **Repetition:** H 0.000 0.000° 0.000° O 0.972 1.571° 0.000° C 1.863 2.342° 0.000° C **3.145 3.145 3.145 3.145 3.145 3.145 3.145 3.145 3.145 3.145** N 4.237 2.389° -0.000° C 4.351 2.686° 0.000° H 5.387 2.751° 0.000° O **3.627 3.627 3.627 3.627 3.627 3.627 3.627** O 2.144 2.693° 3.141°

- **Hallucination:** H 0.000 0.000° 0.000° C 1.117 1.571° -0.000° O 2.022 2.106° -0.000° C 2.211 0.895° -0.000° H 3.096 1.148° 0.000° C 2.789 0.641° **2.157** H 3.166 1.014° -0.965° C 4.055 **4.574** 2.223° H 4.954 0.604° 2.157° H 4.250 0.476° 2.694° C 3.448 0.149° 0.964° H 4.461 0.270° **2.596** C 2.647 0.571° 0.788° H 2.548 0.560° 1.479° C 2.766 2.766 0.769° H 2.596 **4.006 4.302** H 3.722 0.904° 0.936°

In contrast, our approach alleviates both hallucinations and repetitions by incorporating an explicit penalty on invalid molecules during reinforcement learning (RL) (see Eq. 9). Since repetitive token loops and structurally infeasible generations typically result in invalid molecules, penalizing invalidity naturally suppresses these undesired behaviors. This invalidity-aware reinforcement signal constrains the decoding process and substantially reduces the frequency of both hallucinated and repetitive structures, thereby aligning molecular generation more closely with validity requirements.

## F  RETRIEVAL-AUGMENTED PREFIXES: CONSTRUCTION AND SEQUENTIAL IMPLICATIONS

To illustrate how retrieval-augmented prefixes are constructed and to make our approach transparent, we provide several complete examples in this section. Each prefix encodes both the target property value and concise structural statistics—element frequencies and interatomic distance peaks. In this way, a single numeric target is expanded into a structured snapshot of the relevant chemical context. Presenting full prefixes also highlights how their composition varies across properties, making the conditioning process interpretable and showing how POETIC integrates retrieval into language model generation.

Beyond making the conditioning mechanism interpretable, this construction also broadens the model's capacity to generalize. The structural cues carried in the prefix act as transferable priors: even if the target property token is unseen during training, the retrieved statistics bias the model toward chemically plausible regions of the space. Thus, the prefix transforms a brittle scalar condition into a richer, context-aware signal that remains useful across distributions.

This notion of a prefix as a structured and persistent signal becomes even more powerful when paired with sequence-based architectures such as Mamba. Because Mamba generates molecules token by token, the delimiter-bounded prefix tokens do not vanish after initialization; they remain accessible throughout decoding. As a result, the property and structural context encoded in the prefix continually shapes the generative trajectory. What begins as a static condition is effectively turned into a dynamic dialogue between target values and retrieved exemplars, enabling Mamba to maintain both validity and controllability as the sequence unfolds.

**Full serialized examples.** We now present several complete prefixes to illustrate their structure and variability across different target properties.

$\alpha = 86.69$

```
[COND_START] cond value 86.69 [COND_END]
[ELEM_FREQ_START] elems H:0.60,C:0.34,N:0.04,O:0.01 [ELEM_FREQ_END]
[D_PEAK_START] d [2.81,2.97];[2.50,2.66] [D_PEAK_END] [RAG_END]
```

$\alpha = 65.67$

```
[COND_START] cond value 65.67 [COND_END]
[ELEM_FREQ_START] elems H:0.43,C:0.30,N:0.13,O:0.13 [ELEM_FREQ_END]
[D_PEAK_START] d [4.38,4.53];[3.12,3.28] [D_PEAK_END] [RAG_END]
```

$\Delta\epsilon = 0.15$

```
[COND_START] cond value 0.15 [COND_END]
[ELEM_FREQ_START] elems H:0.53,C:0.34,O:0.07,N:0.05 [ELEM_FREQ_END]
[D_PEAK_START] d [4.53,4.69];[0.00,0.16] [D_PEAK_END] [RAG_END]
```

$\epsilon_{\text{HOMO}} = -0.32$

```
[COND_START] cond value -0.32 [COND_END]
[ELEM_FREQ_START] elems H:0.53,C:0.30,N:0.11,O:0.06 [ELEM_FREQ_END]
[D_PEAK_START] d [3.91,4.06];[2.66,2.81] [D_PEAK_END] [RAG_END]
```

Taken together, these examples show that retrieval-augmented prefixes act as a principled bridge between continuous property values and discrete autoregressive generation. In POETIC, this bridge is not a static lookup but a dynamic, sequence-level scaffold: by combining transferable structural priors with explicit token boundaries, it allows Mamba to carry property and structural guidance throughout decoding. This persistent signal helps the model preserve chemical validity while aligning with the target specification, ultimately improving both controllability and generalization within a single, interpretable framework.

# G    VISUALIZATION OF LATENT SPACE ALIGNMENT

To investigate the mechanism behind POETIC's improved controllability, we visualized the latent space embeddings of generated molecules using PCA. As shown in Figure 6, the supervised baseline (MLE) exhibits a disordered embedding space with scattered samples, indicating a lack of correlation between the latent representation and target properties. In contrast, POETIC (after RL) reveals a highly structured manifold with a clear property gradient. This demonstrates that the property-aware RL fine-tuning successfully aligns the model's semantic space with the continuous physical property distribution.

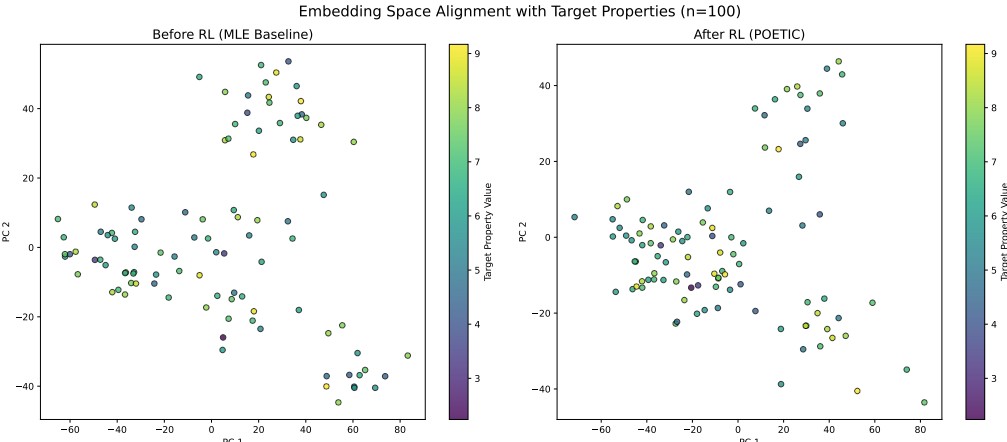

Figure 6: Visualization of latent space embeddings using PCA. The points represent generated molecules colored by their target property values. (Left) The MLE baseline shows a disordered distribution. (Right) POETIC reveals a structured manifold with a clear property gradient, indicating that the model has learned to map continuous physical properties to its internal representation space.

# H    GEOMETRIC DISTRIBUTION ANALYSIS

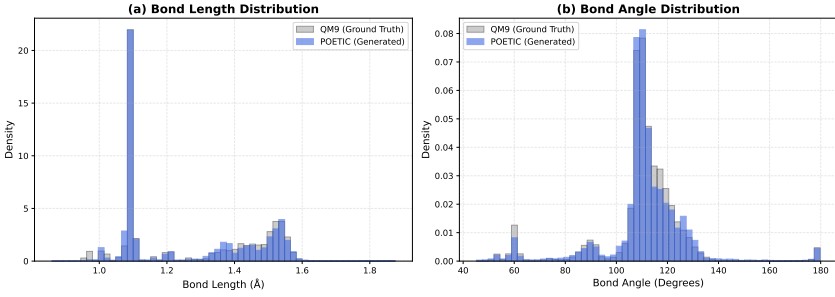

Figure 7: **Comparison of geometric distributions between POETIC and QM9.** The histograms for **(a)** bond lengths and **(b)** bond angles show a high degree of alignment between the generated molecules (blue) and the ground truth (gray), demonstrating the structural realism and geometric fidelity of our approach.

To verify the geometric plausibility of the generated structures, we analyzed the distributions of interatomic bond lengths and bond angles. As shown in Figure 7, the distributions produced by POETIC (blue) exhibit a significant overlap with the ground truth QM9 dataset (gray). The model accurately reproduces the characteristic peaks for both bond lengths and angles, confirming that POETIC successfully captures the underlying physical constraints of stable molecular geometry beyond simple property optimization.

