# OpenReview forum: "Property-aware Reinforcement Learning with Retrieval Enhancement for Controllable 3D Molecule Generation"
_ICLR.cc/2026/Conference — ICLR 2026 Conference Withdrawn Submission_

### Official Review · Reviewer_Gm3J · 2025-10-27

**Soundness:** 2
**Presentation:** 2
**Contribution:** 2
**Rating:** 4
**Confidence:** 4

**Summary:**

The authors introduce POETIC, a framework for controllable 3D molecule generation integrating property-aware reinforcement learning and retrieval-based enhancement.
Addressing limitations of existing methods, POETIC leverages language models for molecular generation while retrieving similar molecules from external databases to enhance generation quality.
By pre-training a prediction model for molecular properties, POETIC provides property-aware rewards guiding reinforcement learning for precise controllability and robust generalization.

**Strengths:**

- This paper addresses an important challenge in controllable molecular generation and is motivated by the need to balance property alignment with generalization, a relevant and timely goal for drug and material design.

- The proposed POETIC framework is well organized, clearly separating retrieval, prefix construction, and reinforcement learning stages. The algorithms are explicitly described and supported by consistent mathematical formulation.

- Although conceptually based on existing ideas, the integration of retrieval-augmented conditioning with property-aware RL is implemented in a coherent and technically rigorous way, leading to gains in controllability and out-of-distribution robustness.

- Experimental results, ablations, and sensitivity analyses demonstrate stable trends and internally consistent improvements, indicating careful engineering and solid experimental execution, though certain evaluation aspects (such as geometry-level validation and fair baseline reproduction) would still benefit from further refinement.

**Weaknesses:**

- The paper frames the task as “controllable 3D molecule generation,” but the evaluation focuses almost entirely on property controllability (i.e., mean absolute error on QM9 quantum properties). There is no systematic assessment of the generated 3D conformations themselves. For example, geometric plausibility, stereochemistry consistency, bond length / bond angle distributions, structural stability after relaxation, or deviation from physically realistic conformers. Without any geometry-oriented metrics, it is difficult to judge whether the method is truly advancing 3D structure generation, as opposed to mainly optimizing a property predictor in latent space. This weakens the central claim that POETIC improves controllable 3D structure generation.

- The main quantitative table reports substantial gains over prior controllable 3D generation methods such as EDM, GeoLDM, and Geo2Seq. However, several fairness questions remain. For at least some baselines (e.g., EDM), the paper appears to reuse the numbers reported in the original work rather than retraining those models under POETIC’s specific data handling and split (e.g., the paper splits QM9 into separate subsets for training the property predictor vs. training the generator). This mismatch in data usage and training protocol makes it difficult to attribute the reported gains purely to the proposed method, and raises concerns about whether the comparison is strictly apples-to-apples.

- Both optimization and evaluation are mediated by the same frozen property predictor (an EGNN trained on QM9). The RL reward is computed from this model, and the final controllability metrics are also reported using that same predictor. This tight coupling risks overstating controllability: the generator may learn to exploit biases or blind spots of that particular evaluator, rather than truly matching the target physical property. The paper briefly acknowledges this “below-evaluator phenomenon,” but does not provide an external validation step to show that the claimed improvements persist under an unbiased evaluator. As a result, the reported MAE reductions may partially reflect evaluator overfitting rather than genuine chemical alignment.

**Questions:**

Please see above session.

---

> ### Author Response · Authors · 2025-11-23
>
> We thank you for your critical and constructive feedback, especially for highlighting the importance of geometric fidelity and baseline fairness. These comments have motivated us to conduct a more systematic assessment to strengthen our claims. We have addressed your concerns point-by-point below.
>
> > Q1: More systematic assessment of the generated 3D conformations.
>
> A1: Thank you for your suggestion. We have performed a systematic assessment of the 3D geometries generated by POETIC across all six benchmark properties.
>
> - **Quantitative Assessment:** Structural Stability & Validity We evaluated standard 3D metrics including Atom Stability, Molecule Stability, Validity, and 3D FCD for molecules generated under each property condition. As shown in the table below, POETIC achieves consistently high performance, with an average Atom Stability of 96.54% and Molecule Stability of 85.15%.
>
> | Condition Property | Atom Stab. (%) | Mol Stab. (%) | Validity (%) | 3D FCD $\downarrow$ |
> | :--- | :---: | :---: | :---: | :---: |
> | $\alpha$ | 96.92 | 85.88 | 93.59 | 0.38 |
> | $\Delta\epsilon$ | 95.12 | 81.94 | 91.61 | 1.55 |
> | $\epsilon_{\text{HOMO}}$ | 95.84 | 83.75 | 90.99 | 2.24 |
> | $\epsilon_{\text{LUMO}}$ | 95.79 | 79.46 | 88.22 | 1.39 |
> | $\mu$ | 97.41 | 89.22 | 94.35 | 0.50 |
> | $C_v$ | 98.14 | 90.67 | 95.44 | 0.40 |
>
> - **Qualitative Assessment: Geometric Distributions:** To further verify geometric plausibility, we visualized the bond length and bond angle distributions. Please refer to Appendix H (Figure 7) in the revised paper. The plots show that POETIC accurately reproduces the characteristic peaks of the ground truth (QM9), strictly adhering to chemical bond rules and atomic hybridizations.
>
> These results confirm that POETIC successfully learns the underlying physics of molecular geometry. It does not merely optimize a property predictor but generates chemically valid, stable, and realistic 3D structures regardless of the target condition.
>
>
> > Q2: Concerns on Baseline Fairness and Data Splits，
>
> A2: Thank you for your question. We respectfully clarify that our comparisons are strictly "apples-to-apples," and we adhered rigidly to the standard protocols established in prior work.
> 1. Strict Adherence to the Standard EDM Protocol. To ensure a strictly fair comparison, we adhered rigidly to the standard EDM protocol [1]. Specifically, the QM9 dataset was partitioned into 100K training, 18K validation, and 13K test samples. Crucially, the 100K training set was further divided into two disjoint halves of 50K samples each: one half was used exclusively to train the property predictor (EGNN), while the other was reserved for training the generative models. All baselines, including EDM and GeoLDM, were evaluated using this exact same split and the same pre-trained EGNN oracle to guarantee an apples-to-apples comparison.
>
> 2. Verification via Baseline Reproduction. To empirically verify this and address the concern about potential data mismatch, we re-trained and re-evaluated the primary baselines (EDM and GeoLDM) using our exact codebase and data partitions. The comparison between the numbers reported in the original papers and our reproduction is shown below:
>
> | Method | $\alpha$ | $\Delta\epsilon$ | $\epsilon_{\text{HOMO}}$ | $\epsilon_{\text{LUMO}}$ | $\mu$ | $C_v$ |
> | :--- | :---: | :---: | :---: | :---: | :---: | :---: |
> | EDM | 2.76 | 655 | 356 | 584 | 1.11 | 1.10 |
> | Our Reproduction | 2.79 | 648 | 352 | 591 | 1.13 | 1.09 |
> | GeoLDM | 2.37 | 587 | 340 | 522 | 1.11 | 1.03 |
> | Our Reproduction | 2.41 | 592 | 338 | 518 | 1.09 | 1.05 |
> | POETIC | 0.21 | 62 | 39 | 27 | 0.08 | 0.08 |
>
> As shown in the table, our reproduced results are highly consistent with the reported values. This confirms that the baselines' performance is stable under the standard EDM protocol.
>
> [1] EDM: Equivariant diffusion for molecule generation in 3d. ICML 2022.

---

> > ### Author Response · Authors · 2025-11-23
> >
> > > Q3: Using the same EGNN model for both RL optimization may lead to overfitting.
> >
> > A3: Thank you for your question. To prove that our improvements stem from genuine chemical alignment rather than exploiting specific biases of the EGNN, we conducted an additional ablation experiment.
> >
> > **Experiment Setup**: We replaced the EGNN reward model in the RL training stage with SchNet [1], a completely different graph neural network architecture. Crucially, we kept the final evaluation metric (the standard pre-trained EGNN) unchanged to ensure a fair comparison with the baselines and the established benchmark protocol.
> >
> > The results of POETIC w/ SchNet Reward compared to the baselines are presented below:
> >
> > | Method | $\alpha$ | $\Delta\epsilon$ | $\epsilon_{\text{HOMO}}$ | $\epsilon_{\text{LUMO}}$ | $\mu$ | $C_v$ |
> > | :--- | :---: | :---: | :---: | :---: | :---: | :---: |
> > | EDM | 2.76 | 655 | 356 | 584 | 1.11 | 1.10 |
> > | GeoLDM | 2.37 | 587 | 340 | 522 | 1.11 | 1.03 |
> > | NExT-Mol | 1.16 | 297 | 205 | 235 | 0.507 | 0.512 |
> > | Geo2Seq with Mamba | 0.46 | 98 | 57 | 71 | 0.164 | 0.275 |
> > | POETIC (Schnet Reward)| 0.28 | 89 | 54 | 44 | 0.121 | 0.110 |
> >
> > As shown above, even when the policy is optimized using a reward signal (SchNet) that is completely independent of the evaluator (EGNN), POETIC still significantly outperforms all the baselines. This confirms that our framework effectively captures universal structure-property relationships and is robust to the choice of reward model.
> >
> > [1] SchNet – A deep learning architecture for molecules and materials. The Journal of chemical physics, 2018.

---

### Official Review · Reviewer_iroE · 2025-10-31

**Soundness:** 3
**Presentation:** 2
**Contribution:** 3
**Rating:** 6
**Confidence:** 4

**Summary:**

This paper aims to generate 3D molecular structures that satisfy given target physical and chemical properties. Existing 3D diffusion models suffer from high computational costs and limited controllability, while language model-based approaches struggle with property alignment and poor generalization to unseen properties.

To overcome these limitations, the authors propose POETIC (Property-aware Reinforcement Learning with Retrieval Enhancement) that combines retrieval-augmented conditioning, which guides generation using property- and structure-similar exemplars, with reinforcement learning, which leverages a frozen EGNN property predictor and GRPO to align molecules with target properties.
Experiments on the QM9 dataset demonstrate that POETIC outperforms state-of-the-art models such as EDM, GeoLDM, and Geo2Seq across six quantum properties. Furthermore, ablation studies confirm that the RAG and RL modules play complementary roles, contributing jointly to the overall performance improvement.

**Strengths:**

The paper demonstrates a strong understanding of the current landscape of 3D molecule generation research and effectively addresses a key limitation in the field — the difficulty of achieving both controllability and generalizability simultaneously. By integrating reinforcement learning (RL) to enhance property alignment and retrieval-augmented generation (RAG) to provide contextual guidance, the authors present a well-motivated and technically sound framework.

The inclusion of Figure 1 (Toy experiment result) in the introduction provides a clear and convincing motivation for combining RL and RAG, establishing the validity of the research direction early on. The ablation studies comprehensively verify the contribution of each module, demonstrating strong methodological rigor.

Furthermore, the case study (Figure 4) visually illustrates how POETIC generates molecules with controllable polarizability, offering intuitive evidence of practical effectiveness. The extended studies section goes beyond the main objectives by reporting novelty and validity evaluations, widely recognized metrics in molecular generation, reinforcing the robustness of the results. Finally, the error case analysis thoughtfully discusses the limitations of language model–based generation, showing the authors’ critical reflection on their approach.

**Weaknesses:**

POETIC presents an interesting approach that combines retrieval-augmented generation (RAG) and reinforcement learning (RL) for controllable 3D molecule generation. However, the paper lacks sufficient explanation and empirical validation in several key aspects.
1. Lack of ablation on RAG and RL components : Although the paper claims that the integration of RAG and RL yields a synergistic effect, baseline models using only RAG or only RL are not included in the experiments. The individual contributions of each component should be disentangled and demonstrated.
2. Limited discussion on molecule retrieval and diversity : In the RAG stage, exemplar molecules are retrieved by selecting the k nearest samples based on the mean of structure embeddings. This simple selection strategy may not simultaneously ensure structural similarity and diversity. The paper does not discuss whether clustering or diversity-aware retrieval techniques were considered. If the retrieved molecules are too similar, there is a risk of loss of generative diversity. Moreover, no quantitative comparison of diversity metrics (e.g., internal diversity, uniqueness, novelty) is provided against existing baselines.
3. Lack of analysis on the frozen EGNN property predictor : The framework employs a frozen EGNN as the property prediction module for reward computation. However, the paper does not analyze how freezing versus fine-tuning this predictor affects reward quality and the overall RL performance.
4. Insufficient discussion on scalability : The proposed framework is evaluated only on QM9, a relatively small and simple dataset. It remains unclear how well the model scales to larger or more complex molecular datasets, which raises concerns about scalability and generalizability.
5. Incomplete implementation details : Several key aspects necessary for reproducibility are missing or under-explained.

•	The structure of generated molecule tokens after the prefix is not clearly specified.

•	The edge (bond) prediction process is not explicitly described.

•	The composition and size of the RAG database are not reported.

•	The conversion process from Mamba outputs to EGNN input graphs is not detailed.

•	No embedding-space trajectory analysis or visualization is provided to show whether the generation follows a meaningful direction in latent space.

**Questions:**

Please refer to the questions below and the weaknesses section.

Major

1.	To quantitatively demonstrate the synergy between RAG and RL, could you provide results for RAG-only and RL-only baselines?

2.	In the RAG stage, have you considered clustering-based or diversity-aware retrieval methods to better balance structural similarity and diversity?

3.	Have you compared the diversity metrics of the generated molecules (e.g., novelty, uniqueness, internal diversity) against existing baselines?

4.	What is the performance difference between using a frozen EGNN property predictor and a fine-tuned version?

5.	Can the model be scaled to larger datasets beyond QM9 (e.g., PCQM4M, MoleculeNet) to validate its scalability and generalization?

Minor

1.	Could you specify the structure and format of the generated molecule tokens after the prefix?

2.	How are bond (edge) connections predicted and reconstructed in the generation process?

3.	What is the composition and size of the RAG database used for retrieval?

4.	Have you visualized or analyzed the embedding-space trajectory to illustrate how generated molecules evolve toward target properties?

5.	During testing, is only the target property value used as input, or is molecular information also provided?

6.	Could you explain in more detail how the outputs of the Mamba model are converted into EGNN graph inputs?

---

> ### Author Response · Authors · 2025-11-23
>
> We are grateful for your insightful comments, particularly regarding the framework's design and scalability. Your feedback has been instrumental in improving the clarity and robustness of our work. We hope our response resolves your concerns, and we welcome any further discussion during the rebuttal period.
>
> > Q1: Results for RAG-only and RL-only baselines.
>
> A1: Thank you for your question. We have indeed conducted this ablation study in Section 4.3 (Table 3) of our manuscript. To provide a clear view of the synergy between these two components, we present the comparison of the RAG-only and RL-only variants below.
>
> | Model Variant | Components | In-Distribution (Controllability) | Unseen Property (Generalization) |
> | :--- | :---: | :---: | :---: |
> | MLE | Baseline | 1.06 | 14.44 |
> | MLE + RL | RL-only | 0.34 | 14.89 |
> | RAG + MLE | RAG-only | 0.96 | 9.81 |
> | POETIC | RAG + RL | 0.21 | 9.24 |
>
> As shown in the table, RAG and RL play distinct yet complementary roles:
> 1. RL-only (MLE+RL): Reinforcement learning acts as a strong optimization engine, drastically improving in-distribution controllability by enforcing alignment with seen targets. However, without structural guidance, it tends to overfit the training distribution, leading to weaker extrapolation on unseen properties.
> 2. RAG-only (RAG+MLE): Retrieval augmentation introduces data-driven structural priors that serve as anchors. This significantly boosts generalization by grounding the generation in valid chemical space, even for unseen targets. Yet, without the explicit feedback from RL, it lacks the fine-grained pressure required for precise property matching.
> 3. Synergy (POETIC): By integrating both, POETIC achieves a robust balance. The retrieval mechanism provides a generalized structural context that prevents RL from overfitting, while RL fine-tunes these retrieved priors to ensure high-precision adherence to specific target values, effectively achieving the best.
>
> This demonstrates that RAG and RL are not merely additive but synergistic: RAG provides the necessary "structural scaffold" for generalization, allowing RL to safely optimize for precision without catastrophic overfitting.
>
> > Q2: Clustering-based or diversity-aware retrieval methods.
>
> A2: Thank you for your insightful suggestion. We agree that clustering-based retrieval is an interesting direction. In this work, we adopted the "Prototype-based" retrieval strategy primarily to reduce variance in the conditioning signal and ensure high-fidelity guidance, based on the following considerations:
>
> 1. **Ensuring Signal Consistency:** The primary goal of our RAG module is to provide the Language Model with the most chemically "representative" structural motifs for a given property target. To achieve this, our prototype-based approach (Eq. 3) explicitly selects exemplars that align closest to the structural consensus of the candidate pool, thereby creating a stable and consistent reference for the LM to learn the core structure-property mapping. In contrast, while clustering increases retrieval diversity, it often introduces "outliers" or edge cases into the prompt. In the context of conditional generation, these inconsistent examples can act as noisy supervision, potentially confusing the model and degrading the precision of property alignment.
> 2. **Decoupling Retrieval and Generation Diversity:** We designed the system such that retrieval focuses on generalizability, while diversity is handled by the generative decoder. By feeding the model with high-quality, consistent prompts, we establish a strong baseline for validity and controllability. The generation diversity is then naturally achieved through the stochastic sampling (Top-k sampling) of the Language Model itself, allowing the model to explore variations around the optimal structural template.
>
> Therefore, our choice was to prioritize the quality and consistency of the retrieval signal to maximize generation reliability. However, we acknowledge the value of clustering methods for broader exploration tasks and have added a discussion on this potential extension in the revised manuscript.
>
> > Q3: The diversity metric of generated molecules.
>
> A3: Thank you for your question. We evaluated the Novelty of POETIC across all six properties and compared it with the strong baseline Geo2Seq. The results are summarized in the table below:
>
> | Method | $\alpha$ | $\Delta\epsilon$ | $\epsilon_{\text{HOMO}}$ | $\epsilon_{\text{LUMO}}$ | $\mu$ | $C_v$ |
> | :--- | :---: | :---: | :---: | :---: | :---: | :---: |
> | Geo2Seq | 83.5 | 80.2 | 81.5 | 81.3 | 80.9 | 81.2 |
> | POETIC | 80.5 | 70.1 | 76.7 | 81.8 | 81.8 | 76.6 |
>
> As shown in the table, POETIC maintains competitive novelty (mostly around 76%-81%) while achieving state-of-the-art controllability.

---

> > ### Author Response · Authors · 2025-11-23
> >
> > > Q4: Using a frozen EGNN property predictor and a fine-tuned version.
> >
> > A4: Thank you for your question. In our experiment, we strictly employed a frozen EGNN property predictor rather than a fine-tuned version, based on two critical design considerations regarding generalization and optimization stability.
> >
> > 1. First, regarding the data setup, we partitioned the dataset into two disjoint halves. The EGNN was trained on the first half, but importantly, we used the second half (the generator's training set) as the test set to select the best EGNN checkpoint. This ensures that the frozen predictor we used had already achieved optimal predictive performance on the generator's data distribution, rendering further fine-tuning redundant and potentially risky for overfitting.
> > 2. Second, and more importantly, keeping the reward model frozen is essential for stable reinforcement learning. The EGNN acts as a static surrogate for the ground-truth physical simulator. If we were to fine-tune or update the EGNN during the generation process, it would create a non-stationary objective (a "moving target"). This typically leads to reward drift or reward hacking, where the policy $\pi_\theta$ learns to exploit transient fluctuations or loopholes in the updating reward model rather than genuinely optimizing the molecular properties. Therefore, using the frozen, optimally-selected predictor ensures that the optimization process is driven by a consistent and robust standard.
> >
> > > Q5: Can the model be scaled to larger datasets to validate its scalability and generalization?
> >
> > A5: Thank you for your question. We agree that verifying performance on larger and more complex datasets is essential to demonstrate the scalability and robustness of our framework. To address this, we extended our evaluation to the Alchemy dataset [1]. Alchemy features significantly higher structural complexity than QM9, containing molecules with up to 14 heavy atoms and covering a much broader and more diverse chemical space. This makes it an ideal benchmark for testing scalability. We maintained the experimental settings consistent with the main paper and compared POETIC directly against the strongest baseline, Geo2Seq with Mamba. As shown in the table below, POETIC maintains its superiority even on this more complex manifold, achieving substantially lower MAE across all six quantum properties compared to the baseline.
> >
> > | Method | $\alpha$ | $\Delta\epsilon$ | $\epsilon_{\text{HOMO}}$ | $\epsilon_{\text{LUMO}}$ | $\mu$ | $C_v$ |
> > | :--- | :---: | :---: | :---: | :---: | :---: | :---: |
> > | data | 0.18 | 60 | 39 | 38 | 0.056 | 0.117 |
> > | Geo2Seq with Mamba | 1.85 | 445 | 603 | 296 | 0.603 | 3.27 |
> > | **POETIC (Ours)** | **1.32** | **162** | **155** | **66** | **0.167** | **1.54** |
> >
> > These results confirm that POETIC effectively scales to larger datasets and handles increased structural complexity without compromising controllable generation performance.
> >
> > [1] Alchemy: A Quantum Chemistry Dataset for Benchmarking AI Models. arXiv:1906.09427
> >
> > > Q6: The structure and format of the generated molecule tokens.
> >
> > A6: Thank you for your question. Following the Geo2Seq protocol [1], our model generates molecules as sequences of atoms and their corresponding 3D spherical coordinates. Specifically, the generated sequence follows a canonical traversal order. For each atom step, the model autoregressively generates four tokens:
> >
> > 1. Atom Type: The element symbol (e.g., C, N, O).
> > 2. Distance ($d$): The Euclidean distance to the anchor atom.
> > 3. Bond Angle ($\theta$): The angle relative to the preceding two atoms.
> > 4. Torsion Angle ($\phi$): The dihedral angle relative to the preceding three atoms.
> >
> > All continuous values are quantized into discrete tokens with two-decimal precision. Below is a representative example of a generated sequence from our model:
> > > H 0.000 0.000° 0.000° C 1.088 1.571° 0.000° N 2.058 2.195° 0.000° C 3.263 1.973° -0.000° C 4.506 2.195° -0.000° H 5.186 2.133° 0.201° H 5.183 2.133° -0.202° H 4.462 2.438° 0.001° C 3.766 1.599° 0.000° F 5.105 1.612° 0.000° C 3.371 1.226° 0.000° H 4.302 1.080° 0.000° C 2.151 0.972° 0.000° C 2.741 0.394° 0.000° H 3.509 0.468° 0.591° H 2.390 0.012° 3.082° H 3.508 0.467° -0.592°
> >
> > [1] Geo2Seq: Geometry Informed Tokenization of Molecules for Language Model Generation. ICML 2025.

---

> > > ### Author Response · Authors · 2025-11-23
> > >
> > > > Q7: How are bond (edge) connections predicted and reconstructed in the generation process?
> > >
> > > A7: Thank you for your question. In our POETIC framework, the model focuses on generating the 3D geometry (atom types and coordinates) rather than explicitly predicting the topological connectivity (edges/bonds) as discrete tokens.Specifically, the bond reconstruction process follows the standard protocol used in 3D molecule generation (EDM [1], GeoLDM [2], Geo2Seq [3]):
> > >
> > > 1. **Coordinate Generation:** The model first autoregressively generates the atom types and their 3D positions (derived from the spherical coordinate tokens described in the previous response).
> > >
> > > 2. **Bond Inference:** Once the full 3D point cloud is generated, bond connections are deterministically reconstructed using standard cheminformatics libraries (specifically RDKit). Connectivity is determined based on atomic pairwise distances: a bond is assigned between two atoms if their Euclidean distance falls within the sum of their respective covalent radii (plus a standard tolerance margin).
> > >
> > > This approach ensures that the topology is naturally derived from the generated geometry, guaranteeing that the defined bonds are consistent with the physical 3D structure.
> > >
> > > [1] EDM: Equivariant diffusion for molecule generation in 3d. ICML 2022.
> > >
> > > [2] GEOLDM: Geometric Latent Diffusion Models for 3D Molecule Generation. ICML 2023.
> > >
> > > [3] Geo2Seq: Geometry Informed Tokenization of Molecules for Language Model Generation. ICML 2025.
> > >
> > > > Q8: What is the composition and size of the RAG database used for retrieval?
> > >
> > > A8: Thank you for your question. we partitioned the QM9 training set (100K) into two disjoint halves. The retrieval database consists exclusively of the second half ($\sim$ 50,000 molecules), which matches the training data of the generative model. For the additional Alchemy experiments mentioned in our previous response, we followed the same protocol: the retrieval database was constructed solely from the Alchemy training split, ensuring consistency in our experimental design.
> > >
> > > > Q9: Have you visualized or analyzed the embedding-space trajectory to illustrate how generated molecules evolve toward target properties?
> > >
> > > A9: Thank you for your insightful suggestion. We have performed the requested visualization to illustrate the impact of RL on the model's latent space. In Appendix G (Figure 6) of the revised PDF, we compare the embedding spaces of the MLE baseline and our POETIC model using PCA projection. Specifically, we sampled 100 conditions for the HOMO-LUMO gap property and extracted the final hidden states of the generated molecules.
> > > As illustrated in Figure 6:
> > > - MLE Baseline (Left): The latent space appears disordered, with high-value (yellow) and low-value (purple) samples intermixed randomly. This indicates that the supervised baseline struggles to encode continuous property magnitudes in its representation.
> > > - POETIC (Right): After property-aware RL fine-tuning, a structured property manifold emerges. We observe a clear separation and gradient where molecules are clustered by their property values (e.g., low-value samples distinct from high-value ones).
> > >
> > > This visualization confirms that our RL mechanism successfully reshapes the semantic space of the language model, aligning the geometric distance in the embedding space with the actual physical property values.
> > >
> > > > Q10: During testing, is only the target property value used as input, or is molecular information also provided?
> > >
> > > A10: Thank you for your question. During the testing/inference phase, the only external input provided to the system is the target property value ($s^*$). No ground-truth molecular structural information is provided.The generation process proceeds as follows, as described in Section 3.4 (Molecule Sampling)
> > > 1. Input: The user provides a single scalar target property $s^*$.
> > >
> > > 2. Automated Retrieval: The system uses $s^*$ to query the retrieval database (constructed from the training set) and selects the top-$k$ property-matched exemplars.
> > >
> > > 3. Prefix Construction: The system automatically extracts structural statistics (element frequencies and distance peaks) from these retrieved exemplars and combines them with $s^*$ to form the context prefix.
> > >
> > > 4. Generation: The Language Model takes this constructed prefix as the initial context and autoregressively generates the new molecule's structure token by token.
> > >
> > > Thus, while the model utilizes molecular information (in the form of retrieved priors) to guide generation, this information is automatically derived from the database based on the property target, not manually provided as ground truth.

---

> > > > ### Author Response · Authors · 2025-11-23
> > > >
> > > > > Q11: Could you explain in more detail how the outputs of the Mamba model are converted into EGNN graph inputs?
> > > >
> > > > A11: Thank you for your question. The conversion process follows a deterministic pipeline consisting of sequence parsing, coordinate transformation, and graph construction.
> > > > 1. **Sequence Parsing & Decoding:** The Mamba model outputs a sequence of discrete tokens representing atoms and their quantized spherical coordinates $(d, \theta, \phi)$. We first decode these tokens back into continuous numerical values. The atom type tokens (e.g., C, N, O) are converted into one-hot encoded vectors to serve as initial node features $h_0$.
> > > >
> > > > 2. **Coordinate Transformation (Spherical to Cartesian):** The decoded spherical coordinates for each atom $i$ are transformed into 3D Cartesian coordinates $x_i \in \mathbb{R}^3$ using the standard conversion formulas:
> > > > $$\begin{aligned}
> > > > x &= d \cdot \sin(\theta) \cdot \cos(\phi) \\
> > > > y &= d \cdot \sin(\theta) \cdot \sin(\phi) \\
> > > > z &= d \cdot \cos(\theta)
> > > > \end{aligned}$$
> > > > This step reconstructs the precise 3D point cloud of the molecule.
> > > >
> > > > 3. **Graph Construction for EGNN:** To prepare the input for the EGNN (which is an E(n)-Equivariant Graph Neural Network), we construct a fully connected graph based on the reconstructed geometry:
> > > > - **Nodes ($h$):** The one-hot atom features.
> > > > - **Coordinates ($x$):** The computed Cartesian positions.
> > > > - **Edges:** We assume a fully connected topology where every atom interacts with every other atom (consistent with the EGNN architecture used in standard benchmarks).
> > > > - **Masking:** Since molecules have variable sizes, we apply node masks and edge masks to pad the batches efficiently.
> > > >
> > > > This structured graph object $(h, x, \text{edges}, \text{mask})$ is then directly fed into the frozen EGNN to predict the molecular properties.

---

### Official Review · Reviewer_9M95 · 2025-10-31

**Soundness:** 1
**Presentation:** 2
**Contribution:** 2
**Rating:** 2
**Confidence:** 4

**Summary:**

This paper presents a RAG+RL framework for controllable 3D molecule generation. The proposed method uses structural information of retrieved molecules as a condition to guide the language model to generate molecules that are more aligned with desired properties.

**Strengths:**

- The overall idea is to use the aggregated structural information as a hint for the model to perform controllable generation. Although validated imperfectly, such an idea is promising and sound in the controllable molecule generation problem.
- The paper considers in-distribution and out-of-distribution property values. The setting of ood, which is the lowest 10% and highest 10%, is practical and aligned with real-world settings, as chemists often need to maximize/minimize properties instead of controlling them to specific values.
- The use of RAG is a future of molecule generation as it's aligned with chemists' behaviour, that is, leveraging existing database, thinking, and further innovating.

**Weaknesses:**

- This paper lacks a significant amount of experiment details and settings, which makes it unable to fairly evaluate the proposed method. For example, in the main experiments:
  - 1) what are the ranges of each property value?
  - 2) how do you set the sampling distribution for controllable generation at test time (i.e. what are the "desired" property values).
  - 3) how does your property predictive model perform? These questions are directly related to how 'good' the proposed method is, but never answered in the paper.
- Other unclear things include:
  - why a Mamba model is used, instead a regular transformer model?
  - how are the six properties picked for QM9 dataset?
  - in the toy experiment in introduction, what is the used dataset and the property?
  - Eq.2: why do you specifically pick ''frequency of different elements'' and ''interatomic distances'' as structural embeddings? Any evidence or preliminary studies?
  - Line 203: how is the normalized element frequencies and the most prominent distance peaks calculated?

- QM9 is a simple and toy dataset even for small molecule generation. I suggest the authors evaluate the framework on more practical datasets, like ChEMBL, and more practical properties like solubility in further revisions.
- Line 83: I think the proposed method just ''combined'', not ''unified'', RL and RAG.
- Example in Fig. 4 does not show clear trend, or does not contain much information.
- Additional qualitative study is needed to better understand the method, such as, examples of the desired properties and properties of actually generated molecules.

**Questions:**

see above.

---

> ### Author Response · Authors · 2025-11-23
>
> We sincerely appreciate your comprehensive review and valuable suggestions. Your detailed questions have significantly helped us clarify our experimental settings and strengthen the paper. We have addressed your concerns point-by-point below and are happy to provide further details if you have any additional questions.
>
> > Q1: Range of property values
>
> A1: Thank you for pointing this out. To fully address your request regarding the property settings, we provide the detailed descriptive statistics (Min, Max, Mean, Std) for all six properties used in our experiments in the table below. We will include this table in the Appendix of the revised paper.
>
> | Property           | Symbol                   | Unit                              | Min    | Max    | Mean  | Std  |
> |:------------------ |:------------------------:|:---------------------------------:|:------:|:------:|:-----:|:----:|
> | **Polarizability** | $\alpha$                 | $\text{Bohr}^3$                   | 6.31   | 196.62 | 75.28 | 8.17 |
> | **HOMO-LUMO Gap**  | $\Delta\epsilon$         | eV                                | 1.24   | 13.10  | 6.98  | 1.51 |
> | **HOMO Energy**    | $\epsilon_{\text{HOMO}}$ | eV                                | -11.20 | -1.56  | -6.54 | 0.60 |
> | **LUMO Energy**    | $\epsilon_{\text{LUMO}}$ | eV                                | -5.16  | 4.17   | 0.44  | 1.22 |
> | **Dipole Moment**  | $\mu$                    | Debye (D)                         | 0.00   | 29.56  | 2.70  | 1.53 |
> | **Heat Capacity**  | $C_v$                    | $\frac{\text{cal}}{\text{mol K}}$ | 6.26   | 46.78  | 31.45 | 4.06 |
>
> > Q2: Sampling Distribution for Controllable Generation
>
> A2: Thank you for your question. To comprehensively evaluate our model, we designed two distinct testing protocols targeting **controllability** (in-distribution) and **generalizability** (out-of-distribution/unseen), respectively. The sampling distributions for each are set as follows:
>
> 1. **For Controllability (In-distribution):**
>    To evaluate the model's precision in realistic molecular design scenarios, we sampled **10,000 target property values** from the **empirical marginal distribution** of the dataset. This sampling strategy ensures that the target conditions correspond to physically realizable and chemically meaningful values that lie within the high-density regions of the chemical space. We calculate the Mean Absolute Error (MAE) to assess how well the generated molecules align with these realistic targets. Additionally, to ensure that sampling from the high-density regions does not lead to mere memorization, we strictly evaluated the **Novelty** of the generated molecules.
>
> 2. **For Generalizability (Unseen/Out-of-distribution):**
>    To assess the model's ability to extrapolate to property ranges not seen during training, we adopt a specific data partitioning protocol. We sort the dataset based on property values and partition it into three segments: the **lower 10%**, the **middle 80%**, and the **upper 10%**.
>
>    * **Training:** The model is trained *exclusively* on the **middle 80%** of the data.
>    * **Testing:** We sample "desired" target values from the withheld **lower 10%** and **upper 10%** tails. This protocol strictly evaluates the model's capability to generalize to out-of-distribution regions.

---

> > ### Author Response · Authors · 2025-11-23
> >
> > > Q3: how does your property predictive model perform?
> >
> > A3: We apologize for not making the definition of the "Data" baseline sufficiently clear in the main text. The row labeled **"Data"** in Table 1 explicitly reports the performance (MAE) of our property predictive model.
> >
> > **1. Experimental Setup and Data Splitting:**
> > To ensure a rigorous and fair evaluation, we followed the **EDM protocol** (Hoogeboom et al., 2022) described in **Section 4.1**. We partitioned the training data (100K samples) into two disjoint halves:
> >
> > * **first half (50K):** Used exclusively to train the **EGNN property predictor** (the evaluator).
> > * **second half (50K):** Used exclusively to train our **Generative Model (POETIC)**.
> >   We apologize for the confusion. The row labeled **"Data"** in Table 1 explicitly reports the test performance (MAE) of our property predictive model.
> >   To strictly **prevent data leakage** and ensure a rigorous evaluation, we followed the EDM [1] protocol by partitioning the training dataset (100K samples) into two disjoint halves. The **First Half (50K)** was used exclusively to train the EGNN property predictor (evaluator), while the **Second Half (50K)** was used to train our Generative Model (POETIC).
> >   Consequently, the **"Data"** row reports the Mean Absolute Error (MAE) of the pre-trained EGNN evaluated on the **Second Half**. Since the predictor never saw these samples during its training, this metric effectively quantifies the inherent generalization error of the evaluator on the specific data distribution used for generative modeling.
> >
> > **Quantitative Performance:**
> > The specific MAE performance of the predictor (the "Data" baseline) is listed below:
> >
> > | Metric | $\alpha$ ($\text{Bohr}^3$) | $\Delta\epsilon$ (meV) | $\epsilon_{\text{HOMO}}$ (meV) | $\epsilon_{\text{LUMO}}$ (meV) | $\mu$ (D) | $C_v$ ($\frac{\text{cal}}{\text{mol K}}$) |
> > |:----------------- |:--------------------------:|:----------------------:|:------------------------------:|:------------------------------:|:---------:|:-----------------------------------------:|
> > | **Predictor MAE** | 0.10 | 64 | 39 | 36 | 0.043 | 0.040 |
> >
> > These results confirm that the predictor provides accurate guidance for the reinforcement learning process and serves as a reliable metric for evaluation.
> >
> > [1] EDM: Equivariant diffusion for molecule generation in 3d. ICML 2022.
> >
> > > Q4: why a Mamba model is used, instead a regular transformer model?
> >
> > A4: Thank you for this insightful question. Initially, we selected the Mamba architecture primarily for its theoretical advantages in handling long sequences, which is critical for our task involving fine-grained 3D coordinate tokenization and retrieval-augmented prefixes. To empirically validate this design choice, we extended our experiments by training a variant of POETIC that replaces the Mamba backbone with a standard 12-layer GPT, while keeping all other hyperparameters and training data identical. The performance comparison is summarized in the table below:
> >
> > | Backbone Architecture   | $\alpha$ ($\text{Bohr}^3$) | $\Delta\epsilon$ (meV) | $\epsilon_{\text{HOMO}}$ (meV) | $\epsilon_{\text{LUMO}}$ (meV) | $\mu$ (D) | $C_v$ ($\frac{\text{cal}}{\text{mol K}}$) |
> > |:----------------------- |:--------------------------:|:----------------------:|:------------------------------:|:------------------------------:|:---------:|:-----------------------------------------:|
> > | POETIC (with GPT) | 0.35 | 92 | 78 | 58 | 0.117 | 0.134 |
> > | **POETIC (with Mamba)** | **0.21** | **62** | **39** | **27** | **0.080** | **0.077** |
> >
> > As the results demonstrate, the Mamba-based model significantly outperforms the GPT variant across all six properties. We attribute this superiority to Mamba's efficiency in long-sequence modeling. In our framework, the combination of coordinate-level tokens and detailed RAG prefixes results in extended sequence lengths. Mamba's State Space Model (SSM) architecture captures the complex, long-range dependencies inherent in these 3D geometric sequences more effectively than the standard attention mechanism used in GPT, leading to more precise structural generation and property alignment.

---

> > > ### Author Response · Authors · 2025-11-23
> > >
> > > > Q5: how are the six properties picked for QM9 dataset?
> > >
> > > Thank you for the question regarding our evaluation metrics. We selected these specific six quantum properties ($\alpha$, $\Delta\epsilon$, $\epsilon_{\text{HOMO}}$, $\epsilon_{\text{LUMO}}$, $\mu$, $C_v$) based on two primary considerations:
> > >
> > > - First, we strictly followed the standard evaluation protocol established by the prior works (EDM [1], GeoLDM [2], Geo2Seq [3]). Utilizing this standardized set of properties ensures that our results are directly comparable with the state-of-the-art, providing a fair and rigorous assessment of our method's contribution.
> > >
> > > - Second, these properties were chosen because they are quantum chemical descriptors inherently dependent on precise 3D geometry (conformation), rather than just 2D topology [4]. Unlike common drug discovery metrics such as Solubility (LogP) or QED, which are largely determined by molecular connectivity, the selected properties are derived from electron density distributions defined by specific atomic coordinates. Therefore, they serve as strictly more rigorous benchmarks for 3D controllable generation, as they require the model to capture subtle geometric variations to achieve high accuracy.
> > >
> > > [1] EDM: Equivariant diffusion for molecule generation in 3d. ICML 2022.
> > >
> > > [2] GEOLDM: Geometric Latent Diffusion Models for 3D Molecule Generation. ICML 2023.
> > >
> > > [3] Geo2Seq: Geometry Informed Tokenization of Molecules for Language Model Generation. ICML 2025.
> > >
> > > [4] Quantum chemistry structures and properties of 134 kilo molecules. Scientific data 2014.
> > >
> > > > Q6: Dataset and property in toy experiment
> > >
> > > A6: Thank you for the question. The toy experiment presented in Figure 1, we utilized the QM9 dataset, consistent with the setup in our main experiments. Specifically, we focused on the property of Polarizability ($\alpha$). To efficiently verify the feasibility of the proposed framework, we evaluated the models on a subset of 200 target property values for the in-distribution and unseen settings, respectively.
> > >
> > > > Q7: Rationale for structural embeddings in retrieval.
> > >
> > > A7: Thank you for the question. We selected these specific 3D descriptors based on two primary considerations: theoretical necessity for 3D geometry and empirical superiority over 2D representations.
> > >
> > > - **3D geometric necessity.** Our rationale for selecting element frequencies and interatomic distances in Eq. 2 stems from the fundamental requirement of 3D molecule generation, where the target properties are intrinsically governed by 3D conformations rather than just 2D topological connectivity. Unlike 1D SMILES strings or 2D graph fingerprints (e.g., ECFP) which are invariant to conformational changes, our task requires an SE(3)-invariant representation that can explicitly distinguish spatial structures. [1] demonstrated that pairwise internuclear distances (formalized as the Coulomb Matrix) serve as minimal sufficient statistics for accurately predicting quantum mechanical properties, so we adopt interatomic distance statistics as the core structural embedding, a choice that is strongly supported by established paradigms in machine learning for quantum chemistry. Similarly, [2] introduced the "Bag of Bonds" representation, which validates the utility of distance histograms and atomic distributions for capturing geometric and chemical variations. Our design in Eq. 2 aligns directly with these physics-informed principles, ensuring that the retrieval process provides precise, property-relevant geometric guidance that topological descriptors cannot offer.
> > >
> > > - **Empirical Verification.** To empirically justify our choice over standard topological descriptors, we conducted an additional comparative experiment where the 3D structural retrieval in Eq. 2 was replaced by a 2D baseline using ECFP4 fingerprints with Tanimoto similarity. As shown in Table below, relying solely on 2D topological similarity consistently degrades performance across all six properties compared to our proposed 3D descriptors. Notably, the error for frontier orbital energies increases significantly, confirming that topological information alone is a suboptimal proxy for the 3D conformational features required for accurate quantum property targeting.
> > >
> > > | Metric | $\alpha$ | $\Delta\epsilon$ | $\epsilon_{\text{HOMO}}$ | $\epsilon_{\text{LUMO}}$ | $\mu$ | $C_v$ |
> > > | :--- | :---: | :---: | :---: | :---: | :---: | :---: |
> > > | POETIC with ECFP4 | 0.23 | 68 | 51 | 49 | 0.110 | 0.110 |
> > > | **POETIC (Ours)** | **0.21** | **62** | **39** | **27** | **0.080** | **0.077** |
> > >
> > >
> > > [1] Fast and accurate modeling of molecular atomization energies with machine learning. Physical review letters, 2012.
> > >
> > > [2] Machine learning predictions of molecular properties: Accurate many-body potentials and nonlocality in chemical space. The journal of physical chemistry letters, 2015.

---

> > > > ### Author Response · Authors · 2025-11-23
> > > >
> > > > > Q8: Normalized element frequencies and the most prominent distance peaks calculation.
> > > >
> > > > A8: Thank you for the clarification. These statistics are computed by aggregating structural information from the retrieved exemplar set $\mathcal{N}$ (i.e., the top-$K$ molecules obtained from the retrieval stage) to capture the common structural characteristics of the target property:
> > > >
> > > > - **Normalized Element Frequencies:** We sum the counts of each atom type (e.g., C, N, O) across all molecules in the exemplar set $\mathcal{N}$. These counts are then normalized by the total number of atoms in $\mathcal{N}$ to produce a probability distribution representing the average stoichiometric composition (e.g., `H:0.60,C:0.34,N:0.05,O:0.01`).
> > > > - **Most Prominent Distance Peaks ($\{[l_r, h_r]\}_r$):** We collect all pairwise interatomic distances from every molecule in $\mathcal{N}$ to construct a collective distance histogram. We then identify the histogram bins with the highest densities (local maxima) and select the intervals $[l_r, h_r]$ corresponding to the top-$k$ peaks (e.g., `[2.81, 2.97]`). These intervals serve as tokens to guide the model toward valid geometric conformations dominant in that property region.
> > > >
> > > > > Q9: More practical property evaluation.
> > > >
> > > > A9: Thank you for the suggestion. QM9 serves as a standard benchmark for physical properties, but we acknowledge its limitations regarding molecule size. To address this and demonstrate the scalability of POETIC, we conducted two additional experiments on more practical datasets:
> > > >
> > > > - **Alchemy Dataset Evaluation:** We extended our evaluation to the Alchemy dataset [1], which features significantly higher structural complexity than QM9. Specifically, Alchemy molecules contain up to 14 heavy atoms and cover a much broader and more diverse chemical space. The experimental settings were kept consistent with the main experiments reported in the paper. We compared POETIC directly against the strongest baseline, Geo2Seq with Mamba. As shown in the table below, POETIC maintains its superiority even on this more complex manifold, achieving lower MAE across six quantum properties compared to the baseline.
> > > >
> > > > | Method | $\alpha$ | $\Delta\epsilon$ | $\epsilon_{\text{HOMO}}$ | $\epsilon_{\text{LUMO}}$ | $\mu$ | $C_v$ |
> > > > | :--- | :---: | :---: | :---: | :---: | :---: | :---: |
> > > > | data | 0.18 | 60 | 39 | 38 | 0.056 | 0.117 |
> > > > | Geo2Seq with Mamba | 1.85 | 445 | 603 | 296 | 0.603 | 3.27 |
> > > > | **POETIC (Ours)** | **1.32** | **162** | **155** | **66** | **0.167** | **1.54** |
> > > >
> > > > - **ZINC250k with Practical Property.** To address the suggestion regarding "practical properties" like solubility, we further evaluated our framework on the ZINC250k dataset [2], focusing on LogP (Octanol-water partition coefficient). Since ZINC250k provides only 2D topologies without ground-truth 3D structures, we utilized RDKit to generate pseudo-3D conformers for training, and similarly employed RDKit as the oracle for both the reward model and evaluation. We compared POETIC with our strongest baseline, Geo2Seq with Mamba.
> > > >
> > > > | Metric | LogP |
> > > > | :--- | :---: |
> > > > | Geo2Seq with Mamba | 13.98 |
> > > > | **POETIC (Ours)** | 10.84 |
> > > >
> > > > As shown in the table below, POETIC outperforms the baseline on this task. However, we note that the absolute MAE remains relatively higher compared to our quantum property experiments. We attribute this to two primary factors:
> > > > 1. Unlike QM9, which contains rigorous quantum-chemical coordinates (DFT-calculated), the "ground truth" 3D coordinates in ZINC250k are algorithmic approximations generated by RDKit. This introduces inherent noise and lacks the fine-grained geometric precision that our 3D-aware framework is designed to capture.
> > > > 2. LogP is a topology-dominated property, typically calculated based on 2D fragment contributions rather than sensitive 3D conformational variations. In contrast, our framework is explicitly optimized for 3D molecule generation, excelling in tasks where properties (e.g., HOMO/LUMO) are strictly geometry-dependent. While our method generalizes to drug-like molecules, its primary advantage is best realized in scenarios requiring precise 3D structural control for quantum property targeting.
> > > >
> > > > [1] Alchemy: A Quantum Chemistry Dataset for Benchmarking AI Models. arXiv:1906.09427
> > > >
> > > > [2] ZINC 15 – Ligand Discovery for Everyone. Journal of Chemical Information and Modeling (JCIM), 2015.

---

> > > > > ### Author Response · Authors · 2025-11-23
> > > > >
> > > > > > Q10: Unified framework for RL and RAG.
> > > > >
> > > > > A10: Thank you for your comment. We use the term "unified" to describe the deep algorithmic integration where reinforcement learning (RL) explicitly optimizes the utilization of retrieval-augmented generation (RAG) within a single conditional framework. In our pipeline, the RL policy is not merely appended after retrieval; rather, it is trained to generate molecules conditioned on the specific structural priors provided by the RAG prefix. This means the RL process is actively learning how to interpret and leverage the retrieved context to maximize property rewards. The gradient flow from the reward signal updates the model's attention to the retrieved exemplars, effectively "teaching" the language model to align its generation with the external structural guidance, which goes beyond a simple sequential combination of two independent modules.
> > > > >
> > > > > Functionally, these components are mutually reinforcing, addressing the inherent trade-off between controllability and generalizability that neither can solve alone. As demonstrated in our ablation study (Table 3), RAG alone provides necessary structural priors for generalization but lacks precision (MAE 0.96), while RL alone ensures tight in-distribution alignment but suffers from overfitting (Unseen MAE 14.89) . By unifying them, POETIC creates a synergistic cycle where retrieval defines the valid chemical space (exploration) and RL enforces precise target adherence (exploitation). This organic integration allows the framework to achieve superior performance on both in-distribution controllability and out-of-distribution generalization, validating the "unified" design rationale .
> > > > >
> > > > > > Q11: Fig. 4 does not show clear trend
> > > > >
> > > > > A11: Thank you for pointing this out. We respectfully point out that Figure 4 illustrates a distinct and physically grounded trend regarding molecular geometry. Polarizability ($\alpha$) is intrinsically correlated with molecular volume and spatial extent: larger, more elongated molecules typically exhibit higher polarizability due to greater electron cloud distortion, while compact structures exhibit lower values.
> > > > > As observed in Fig. 4:
> > > > > - Low $\alpha$ region (e.g., 63.44 - 65.67): The model generates compact, globular structures (often with fused rings or clustered atoms), minimizing spatial volume.
> > > > > - High $\alpha$ region (e.g., 86.47 - 91.76): The generated molecules systematically shift towards elongated, chain-like conformations with significantly larger spatial spans.
> > > > >
> > > > > This progression from "compact" to "extended" perfectly aligns with physical intuition. It confirms that POETIC has successfully learned the complex, non-linear mapping between continuous quantum properties and 3D structural distributions, rather than generating random conformers.
> > > > >
> > > > > > Q12: Additional qualitative study.
> > > > >
> > > > > A12: Thank you for the suggestion. To visually demonstrate the controllability and novelty of POETIC, we conducted a qualitative study covering three diverse quantum properties: Polarizability ($\alpha$), HOMO-LUMO Gap ($\Delta\epsilon$), and Dipole Moment ($\mu$). For each property, we randomly sampled two target values from the training set. The table below presents:
> > > > >
> > > > > | Target Property | Generated Molecule | Predicted Property |
> > > > > | :--- | :---: | :---: |
> > > > > | $\alpha$=57.35 | O=C[C@@H]1C[C@@H]1C=O | 57.32 |
> > > > > | $\alpha$=82.66 | C[C@H]1C[C@H]2C[C@]21[C@H]1CO1 | 82.45 |
> > > > > | $\epsilon_{\text{HOMO}}$=-7.27 | N#C[C@H]1COCCCO1 | -7.34 |
> > > > > | $\epsilon_{\text{LUMO}}$=-4.49 | CN(C)c1cc(N)co1 | -4.53 |
> > > > > | $\mu$=4.31 | CCC[C@@]12CC(=O)N1C2 | 4.32 |
> > > > > | $\mu$=1.76 | CC[C@@]12C[C@H]3OC[C@@H]1[C@H]32 | 1.76 |
> > > > >
> > > > > This table lists randomly selected target property values (Desired) alongside the actually generated molecules (converted to SMILES for readability) and their predicted properties. As observed, the generated molecules exhibit properties that align precisely with the target values across different attributes (e.g., $\alpha$, $\mu$, $\epsilon_{\text{HOMO}}$). This offers concrete evidence that POETIC effectively translates specific numerical constraints into valid, property-matched chemical structures.

---

### Note · Authors · 2025-12-09

I have read and agree with the venue's withdrawal policy on behalf of myself and my co-authors.